# Zero-Shot Context Generalization in Reinforcement Learning from Few Training Contexts

*James Chapman
Department of Mathematics
University of California, Los Angeles
Los Angeles, CA 90095
chapman20j@math.ucla.edu

*Kedar Karhadkar
Department of Mathematics
University of California, Los Angeles
Los Angeles, CA 90095
kedar@math.ucla.edu

Guido Montúfar
Departments of Mathematics and of Statistics & Data Science
University of California, Los Angeles
Los Angeles, CA 90095
montufar@math.ucla.edu

## Abstract

Deep reinforcement learning (DRL) has achieved remarkable success across multiple domains, including competitive games, natural language processing, and robotics. Despite these advancements, policies trained via DRL often struggle to generalize to evaluation environments with different parameters. This challenge is typically addressed by training with multiple contexts and/or by leveraging additional structure in the problem. However, obtaining sufficient training data across diverse contexts can be impractical in real-world applications. In this work, we consider contextual Markov decision processes (CMDPs) with transition and reward functions that exhibit regularity in context parameters. We introduce the context-enhanced Bellman equation (CEBE) to improve generalization when training on a single context. We prove both analytically and empirically that the CEBE yields a first-order approximation to the Q-function trained across multiple contexts. We then derive context sample enhancement (CSE) as an efficient data augmentation method for approximating the CEBE in deterministic control environments. We numerically validate the performance of CSE in simulation environments, showcasing its potential to improve generalization in DRL.[1]

## 1 Introduction

Deep reinforcement learning has been deployed successfully in many problems requiring decision making and multi-step optimization. Despite these successes, reinforcement learning still struggles to perform well at test time in real-world scenarios. This is a well documented phenomenon with multiple causes (e.g., sim-to-real distribution shift, limited contextual data) (Zhao et al., 2020; Ghosh et al., 2021; Kirk et al., 2023). This places out-of-distribution generalization as a fundamental challenge of reinforcement learning.

One approach to address generalization in reinforcement learning is to use continual learning in which some training occurs at deployment in the testing environment (Khetarpal et al., 2022; Wang et al., 2023). However, this is not always feasible or desirable, as, for instance, safety considerations

---

*Equal Contribution.
[1]Code: https://github.com/chapman20j/ZeroShotGeneralization-CMDPs.

may prevent early agent deployment, and online adaptation may be prohibitively expensive (Gu et al., 2024; Groshev et al., 2023). Instead, we need to ensure that the agent has good *zero-shot* generalization from only the training data. Another approach is to create many training environments to prevent over-optimization on specific context parameters (e.g., friction coefficients) as in Meta-RL and domain randomization (Beck et al., 2025; Chen et al., 2022). This is commonly used in robotics where sim-to-real pipelines are necessary, and it has been found to produce policies that are more robust to parameter uncertainty (e.g., less sensitive to distribution shift) (Tang et al., 2024; Zhao et al., 2020). However, the environment construction may be prohibitively expensive. For example, designing and building complex robots requires substantial engineering labor and expensive hardware.

Without sufficient structure and prior information, zero-shot generalization from only a few contexts is an impossible problem. However, in many domains we have knowledge of the underlying dynamics of the environment. For example, when solving control problems we may have equations for the dynamics and only uncertainty in the parameters of the equations (Kay, 2013). In such cases it may be possible to include knowledge about the general form of the dynamics into the architecture and/or training algorithm without ever sampling from these unseen contexts. Some works build architectures which try to capture such biases in the underlying dynamics (Wang et al., 2018; Kurin et al., 2021; Huang et al., 2020). Although this approach tends to improve training performance, the generalization benefits have remained limited. The authors of these works note that the variance on the evaluation set is still relatively large. Attaining consistent improvements in out-of-distribution contexts typically requires developing complex architectures which are specifically tailored to the problem at hand (Hong et al., 2022; Chen et al., 2023; Xiong et al., 2023; Li et al., 2024). This is a valuable approach, but is challenging to scale to larger sets of environments as it requires domain specific information.

## 1.1 Our Contributions

In this paper we develop an approach to incorporate structure across contexts into the training algorithm in order to improve generalization performance on previously unseen contexts. We introduce the *context-enhanced Bellman equation* (CEBE), which is the Bellman equation of a contextual Markov decision process (CMDP) with transition and reward functions linearized about a particular training context. This allows us to estimate data from the CMDP in a neighborhood of the training context. We also derive *context sample enhancement* (CSE) as an efficient method to generate data from nearby contexts in the linearized CMDP that we can then use to optimize the CEBE in deterministic control environments. This allows us to enhance the data sampled from a single training context to effectively train on nearby contexts. Our contributions are as follows:

1. We provide a theoretical analysis of context perturbations in contextual Markov decision processes (CMDPs), demonstrating the viability of our proposed CEBE method for improving out-of-distribution generalization.

2. We derive CSE as an efficient data augmentation method for deep reinforcement learning (DRL), enabling more robust policy learning when learning from samples generated in the training context of the CMDP.

3. We perform experiments with CEBE and CSE on a variety of RL environments that exhibit regularity in the transition and reward functions with respect to the context parameters, which demonstrate the utility of our methods to improve out-of-distribution generalization in context-limited training scenarios.

This paper is organized as follows. Section 2 provides an overview of relevant background. Section 3 introduces CEBE and our theoretical analysis of CEBE. Section 4 introduces CSE and provides an algorithm for data augmentation with CSE. Section 5 shows our experimental evaluation of CEBE and CSE. Section 6 offers a discussion and concluding remarks.

## 1.2 Related Works

Here we briefly discuss some of the most closely related works and defer a more extensive overview of related works to Appendix A. A related strategy for introducing knowledge of the dynamics into RL training has been considered in previous work focusing on the action space. Qiao et al. (2021) considered gradients with respect to actions and introduced *sample enhancement* and *policy enhancement* to augment samples from the replay buffer. We instead focus on the context variable towards

addressing the problem of context generalization, for which we develop theory and experiments. In the stochastic setting, our proposed CEBE can be estimated using importance sampling. This would then be related to Tirinzoni et al. (2018), who considered importance sampling for improving out-of-distribution zero-shot generalization in RL. That work considers exact transitions and rewards and must therefore consider information from nearby contexts.

Modi et al. (2018) assume continuity in the context space and propose a covering algorithm based on zeroth order approximation of the policy in context space to derive PAC bounds. This requires access to many contexts, but we note that our first-order approximation provides better coverage of the context space. Malik et al. (2021) consider CMDPs which have transitions and rewards that are uniformly close in a neighborhood to provide lower bounds on the number of context queries required for generalization. Our setting differs because we only sample one context, we consider first-order information, and we obtain approximation results under more relaxed assumptions on the state and action spaces. We also provide a practical algorithm for achieving generalization comparable to domain randomization. Levy and Mansour (2022) derive polynomial sample complexity bounds when one has access to an ERM oracle and is able to sample from nearby contexts. This differs from our work as we seek to approximate the dynamics and reward functions of nearby contexts without sampling.

## 2 Contextual Markov Decision Processes

Sequential decision making is often modeled as a Markov decision process (MDP), and deep reinforcement learning is employed to solve complex MDPs. However, the MDP framework does not directly account for changes in the underlying context, and policies trained via DRL tend to overfit to context parameters. Contextual MDPs (CMDPs) extend MDPs by including context parameters explicitly in the MDP dynamics and rewards, which allows one to more directly account for their impact. The context parameters can include a variety of aspects, particularly parameters that influence the transition dynamics and parameters that specify a task by modifying the reward function.

We recall the relevant definitions and notation, consistent with Beukman et al. (2023), that we use to formulate our results below. A CMDP is defined as

$$\mathcal{M} = (\mathcal{C}, \mathcal{S}, \mathcal{A}, \mathcal{M}', \gamma), \tag{1}$$

where $\mathcal{C}$ is the context space, $\mathcal{S}$ is the state space, $\mathcal{A}$ is the action space, and $\gamma \in [0,1]$ is the discount factor. Here $\mathcal{M}'$ is a function that takes any context $c \in \mathcal{C}$ to a corresponding MDP

$$\mathcal{M}'(c) = (\mathcal{S}, \mathcal{A}, \mathcal{T}^c, R^c, \gamma), \tag{2}$$

where $\mathcal{T}^c$ and $R^c$ are the transition and reward functions of the context-$c$ MDP. For a specific $c \in \mathcal{C}$, $\mathcal{T}^c$ is a map $\mathcal{S} \times \mathcal{A} \to \Delta(\mathcal{S})$, where $\Delta(\mathcal{S})$ is the space of probability distributions on $\mathcal{S}$. In general the state and action spaces could depend on $c$, but we restrict to a common state and action space in this paper. We consider policies of the form $\pi : \mathcal{C} \times \mathcal{S} \to \Delta(\mathcal{A})$. Given an initial state distribution, we define the expected discounted return in context $c$ when using policy $\pi(\bullet | c, s)$ as

$$J(\pi, c) = \mathbb{E}_{s_0, \pi, \mathcal{T}^c} \sum_{t \geq 0} \gamma^t R_t^c. \tag{3}$$

For a set of contexts $U \subseteq \mathcal{C}$, we say that a policy $\pi$ is $(U, \epsilon)$-*optimal* if

$$J(\pi, c) \geq J(\rho, c) - \epsilon, \tag{4}$$

for all $c \in U$ and all policies $\rho : \mathcal{S} \to \Delta(\mathcal{A})$. Given a test context distribution $\mathcal{D}^{\text{test}}$ with support $S \subseteq \mathcal{C}$, the objective is to find a $(S, \epsilon)$-optimal policy. In this paper, we consider the out-of-distribution setting where the train and test context distributions, $\mathcal{D}^{\text{train}}$ and $\mathcal{D}^{\text{test}}$, have different supports.

**Structural Assumptions** We will consider the setting where the agent must generalize from a single context and in particular the policy must generalize to contexts that are unseen during training. For this to be possible we must add structure to the CMDP. We place the following assumptions:

1. The state space $\mathcal{S}$ and action space $\mathcal{A}$ are metric spaces. The context space $\mathcal{C}$ is an open convex subset of a Banach space.

2. The reward function $R^c : \mathcal{S} \times \mathcal{A} \times \mathcal{S} \to \mathbb{R}$ is deterministic.

3. The partial derivatives $\partial_c R^c$ and $\partial_c \mathcal{T}^c$ exist and are Lipschitz.

4. The context $c \in \mathcal{C}$ is fully observable and fixed over an episode.

5. When making a transition $(s, a) \mapsto s'$, the agent observes $\partial_c R^c(s, a, s')$ and $\partial_c \mathcal{T}^c(s'|s, a)$.

Settings of this form commonly appear in physical environments where the underlying dynamics are determined by a system of differential equations with physical parameters (e.g., friction coefficient, elastic modulus, mass) that may be modeled in terms of a context $c$.

# 3 Perturbative Theory of CMDPs

## 3.1 Context-Enhanced Bellman Equation (CEBE)

$$Q_{\mathrm{BE}}(s, a, c) = \mathbb{E}_{s' \sim \mathcal{T}^c(s,a)} \left( R^c + \gamma \mathbb{E}_{a' \sim \pi(s',c)} Q_{\mathrm{BE}}(s', a', c) \right) \tag{5}$$

is a fundamental equation in RL which is used to train the $Q$-function. However, the transitions and rewards may not be known in all contexts. We fix a base training context $c_0$ and suppose that we have access to approximate transition and reward functions $\mathcal{T}_{\mathrm{CE}}$ and $R_{\mathrm{CE}}$ satisfying:

$$\mathcal{T}_{\mathrm{CE}}^{c_0} = \mathcal{T}^{c_0}, \quad R_{\mathrm{CE}}^{c_0} = R^{c_0}, \quad \mathcal{T}_{\mathrm{CE}}^c \approx \mathcal{T}^c, \quad R_{\mathrm{CE}}^c \approx R^c$$

for all $c$ in a neighborhood of $c_0$. We refer to such approximations as the *context-enhanced transitions* and *context-enhanced rewards*, respectively. We define the context-enhanced Bellman equation (CEBE) as the Bellman equation of the approximate CMDP:

$$Q_{\mathrm{CE}}(s, a, c) = \mathbb{E}_{s' \sim \mathcal{T}_{\mathrm{CE}}^c(s,a)} \left( R_{\mathrm{CE}}^c + \gamma \mathbb{E}_{a' \sim \pi(s',c)} Q_{\mathrm{CE}}(s', a', c) \right). \tag{6}$$

Note that at $c = c_0$ this coincides with the original Bellman equation. While our theoretical results further below apply in more general settings, in this paper we will let $R_{\mathrm{CE}}$ be the Taylor approximation of $R$ about $c_0$ and primarily consider the following two cases with corresponding definitions of $\mathcal{T}_{\mathrm{CE}}$.

**Deterministic Transitions** For the setting of deterministic transitions, there exists a function $f^c(s, a)$ such that $\mathcal{T}^c(s, a) = \delta_{f^c(s,a)}$ is a Dirac delta distribution for each $s, a, c$. In this case, we define the context-enhanced transition and reward functions as

$$\mathcal{T}_{\mathrm{CE}}^c(s, a) = \delta_{f^{c_0}(s,a) + \partial_c f^{c_0}(s,a)(c-c_0)} \tag{7}$$

$$R_{\mathrm{CE}}^c = R^{c_0} + \partial_c R^{c_0} \cdot (c - c_0) + \partial_{s'} R^{c_0} \partial_c \mathcal{T}^{c_0} \cdot (c - c_0). \tag{8}$$

This is particularly useful in the online setting as we can simply estimate the CEBE from samples. Here we additionally assume that the reward function is differentiable in the next state to compute its linear approximation (i.e., $\partial_{s'} R^c$ exists and is Lipschitz).

**Transitions with Differentiable Measure** If $c \mapsto \mathcal{T}^c(s, a)$ is a differentiable map, then we can define the context-enhanced transition and reward functions by

$$\mathcal{T}_{\mathrm{CE}}^c(s, a) = P(\mathcal{T}^{c_0}(s, a) + \partial_c \mathcal{T}^{c_0}(s, a) \cdot (c - c_0)) \tag{9}$$

$$R_{\mathrm{CE}}^c = R^{c_0} + \partial_c R^{c_0} \cdot (c - c_0) \tag{10}$$

where $P(\mu) = \frac{\mu^+}{\|\mu^+\|}$ denotes projection onto the probability simplex and $\mu^+$ denotes the positive part of the signed measure $\mu$. This projection is required since the linear approximation may produce a measure that is negative in places. If $\Delta c = c - c_0$ is sufficiently small, the projection is well-defined and sends the linear approximation to a probability measure.

## 3.2 CEBE Approximation of the Bellman Equation

In this subsection, we prove that the context-enhanced transitions and rewards we introduced above lead to a $Q$-function $Q_{\mathrm{CE}}$ which is close to $Q_{\mathrm{BE}}$. For our approximation results, we introduce the following notation on spaces of functions and measures. If $X$ and $Y$ are metric spaces and

$f : X \to Y$ is Lipschitz, we denote its Lipschitz constant by $L_f$. If $X$ is a metric space, we let $\mathrm{Lip}(X)$ denote the Banach space of Lipschitz functions $X \to \mathbb{R}$, equipped with the metric

$$\|f\|_{\mathrm{Lip}(X)} = \max \left( \sup_{x \in X} |f(x)|, \sup_{\substack{x,y \in X \\ x \neq y}} \frac{|f(x) - f(y)|}{d(x,y)} \right).$$

For a metric space $X$, let $\mathrm{Meas}(X)$ denote the space of finite signed Borel measures on $X$ equipped with the total variation norm. If $X$ is a metric space and $p \in [1, \infty]$, let $\mathcal{W}_p(X)$ denote the space of probability measures on $X$, equipped with the Wasserstein distance

$$W_p(\mu, \nu) = \inf_{\gamma \in \Pi(\mu,\nu)} \left( \int d(x,y)^p d\gamma(x,y) \right)^{1/p}.$$

Here $\Pi(\mu, \nu)$ denotes the space of probability distributions on $X \times X$ which have marginal distributions $\mu$ and $\nu$. For $p = \infty$, we define $W_\infty(\mu, \nu) = \lim_{p \to \infty} W_p(\mu, \nu)$.

Now we present our main result, which establishes that $Q$-functions are stable under small perturbations to the transition dynamics and rewards.

**Theorem 1** (($\mathcal{T}, R$)-stability of the $Q$-function). *Let $R^{(1)}, R^{(2)} \in \mathrm{Lip}(\mathcal{S} \times \mathcal{A})$ be reward functions with $\|R^{(1)} - R^{(2)}\|_\infty \leq \delta_R$. Let $\mathcal{T}^{(1)}, \mathcal{T}^{(2)} : \mathcal{S} \times \mathcal{A} \to \mathcal{W}_p(\mathcal{S})$ be transition functions with*

$$\sup_{(s,a) \in \mathcal{S} \times \mathcal{A}} W_p(\mathcal{T}^{(1)}(s,a), \mathcal{T}^{(2)}(s,a)) \leq \delta_T,$$

*and let $\pi : \mathcal{C} \times \mathcal{S} \to \mathcal{W}_p(\mathcal{A})$ be a Lipschitz policy. Let $\gamma \in (0,1)$ be a discount factor with $\gamma < \frac{1}{\max(L_{\mathcal{T}^{(1)}}, L_{\mathcal{T}^{(2)}})(1+L_\pi)}$. Let $Q^{(1)}$ and $Q^{(2)}$ denote solutions of the Bellman equation for $(\mathcal{T}^{(1)}, R^{(1)}, \gamma)$ and $(\mathcal{T}^{(2)}, R^{(2)}, \gamma)$ respectively. Then*

$$\|Q^{(1)} - Q^{(2)}\|_\infty \leq \frac{1}{1-\gamma} \left( \delta_R + \frac{\gamma(1+L_\pi)\delta_T \|R^{(2)}\|_{\mathrm{Lip}(\mathcal{S} \times \mathcal{A})}}{1 - \gamma \max(1, L_{\mathcal{T}^{(2)}}(1+L_\pi))} \right).$$

*Proof sketch.* Using that $Q^{(1)}$ and $Q^{(2)}$ are solutions to their respective Bellman equations, we can decompose

$$Q^{(1)} - Q^{(2)} = (R^{(1)} - R^{(2)}) + (\gamma A^{(1)} Q^{(1)} - \gamma A^{(1)} Q^{(2)}) + (\gamma A^{(1)} Q^{(2)} - \gamma A^{(2)} Q^{(2)}),$$

where $A^{(1)}$ and $A^{(2)}$ are transition operators for the dynamics in the two MDPs under policy $\pi$. The first term is small if $R^{(1)}$ and $R^{(2)}$ are sufficiently close. The second term can be bounded in terms of the norms of $A^{(1)}$ and $Q^{(1)} - Q^{(2)}$. The third term can be bounded by showing that $A^{(1)}$ and $A^{(2)}$ are close, and showing that the learned function $Q^{(2)}$ is sufficiently smooth. Combining these bounds and using a triangle inequality, we obtain the desired bound on $\|Q^{(1)} - Q^{(2)}\|_\infty$. $\square$

The full proof for Theorem 1 is contained in Appendix B.2. Next we apply the theorem in the setting where $Q^{(1)}$ and $Q^{(2)}$ represent different contexts of a CMDP. We first consider the case where the transition dynamics are deterministic and we use context enhancement as in (7) and (8). For this case, we assume that $\mathcal{S}$ and $\mathcal{A}$ are Banach spaces, so we can differentiate with respect to states and actions.

**Theorem 2** (Deterministic CEBE is first-order accurate). *Consider a Lipschitz policy $\pi : \mathcal{S} \times \mathcal{C} \to \mathcal{W}_p(\mathcal{A})$ and deterministic $\mathcal{T}$. Suppose that the discount factor $\gamma$ satisfies*

$$\gamma < \frac{1}{(\|D\mathcal{T}\|_\infty + \|D^2\mathcal{T}\|_\infty \|c - c_0\|)(1+L_\pi)}.$$

*Let the context-enhanced transitions and rewards be defined by (7) and (8), respectively. Then*

$$\|Q_{\mathrm{CE}}^c - Q_{\mathrm{BE}}^c\|_\infty \leq \frac{\|c - c_0\|^2}{1-\gamma} \left( \|D^2 R\|_\infty + \frac{\gamma(1+L_\pi)\|D^2\mathcal{T}\|_\infty(\|R\|_\infty + \|DR\|_\infty)}{1 - \gamma \max(1, \|D\mathcal{T}\|_\infty(1+L_\pi))} \right).$$

This establishes that for a fixed policy $\pi$, the $Q$-function of the CEBE approximates the $Q$-function of the original CMDP with $O(\|c - c_0\|^2)$ error, provided that the transition and reward functions are sufficiently smooth. We prove Theorem 2 in Appendix B.3. Next, we establish an analogue of Theorem 2 for the case where the transition map is stochastic.

**Theorem 3** (Stochastic CEBE is first-order accurate). *Consider a Lipschitz policy $\pi : \mathcal{S} \times \mathcal{C} \rightarrow \mathcal{W}_p(\mathcal{A})$ and stochastic $\mathcal{T}$. Let $c_0, c \in \mathcal{C}$ with $\|c - c_0\| < \|\partial_c^2 \mathcal{T}\|_\infty^{-1/2}$ and*

$$\gamma < \frac{1}{4 \operatorname{diam}(\mathcal{S})(L_\mathcal{T} + \|c - c_0\| L_{\partial_c \mathcal{T}})(1 + L_\pi)}.$$

*Let the context-enhanced transitions and rewards be defined by* (9) *and* (10)*, respectively. Then*

$$\|Q_{\mathrm{CE}}^c - Q_{\mathrm{BE}}^c\|_\infty \leq \frac{\|c - c_0\|^2}{1 - \gamma} \left( \|\partial_c^2 R\|_\infty + \frac{3\gamma \operatorname{diam}(\mathcal{S})(1 + L_\pi)\|R\|_{\mathrm{Lip}(\mathcal{S} \times \mathcal{A} \times \mathcal{C})}\|\partial_c^2 \mathcal{T}\|_\infty}{1 - \gamma \max(1, \operatorname{diam}(\mathcal{S})L_\mathcal{T}(1 + L_\pi))} \right).$$

*Here $L_\mathcal{T}$ denotes the Lipschitz constant of $\mathcal{T}$ as a map $\mathcal{S} \times \mathcal{A} \times \mathcal{C} \rightarrow \operatorname{Meas}(\mathcal{S})$.*

We prove Theorem 3 in Appendix B.4. The above two theorems show that training with CEBE does not incur too much approximation error versus training with BE in a neighborhood of the training context. In particular, we can use CEBE to approximately solve the Bellman equation. The following theorem complements these results by showing that a policy optimized using CEBE will also be close to optimal on the original Bellman equation.

**Theorem 4.** *Let $\pi_{\mathrm{CE}}$ be an $L$-Lipschitz policy which is $(\mathcal{C}, \epsilon)$-optimal with respect to the CEBE, and suppose that there exists an $L$-Lipschitz policy $\pi_{\mathrm{BE}}$ which is $(\mathcal{C}, \epsilon)$-optimal with respect to the BE. Suppose that*

$$\|Q_{\mathrm{CE}}^c(\bullet, \bullet; \pi) - Q_{\mathrm{BE}}^c(\bullet, \bullet; \pi)\|_\infty < \delta$$

*for all $c \in \mathcal{C}$ and all $L$-Lipschitz policies $\pi$, and suppose that the reward function $R$ is bounded. Then $\pi_{\mathrm{CE}}$ is $(\mathcal{C}, 2\delta + 2\epsilon)$-optimal with respect to the Bellman equation.*

The proof of the above theorem uses the observation that if two functions are uniformly close, then a good optimizer of one is also a good optimizer of the other. We provide a proof in Appendix B.5.

## 4 Context Sample Enhancement

In this section, we show how one can estimate the CEBE from samples in the deterministic-transitions case. Suppose we are given a dataset $\mathcal{D}$ consisting of samples of the form $(s, a, r, s')$. We introduce the following context sample enhancement CSE procedure, which takes a sample and a context perturbation $\Delta c = c - c_0$ as inputs and returns a context-enhanced sample:

$$\mathrm{CSE}((s, a, r, s'), \Delta c) = (r + \partial_c R^{c_0}(s, a, s')\Delta c + \partial_{s'} R^{c_0}(s, a, s')\partial_c \mathcal{T}^{c_0}\Delta c, s' + \partial_c \mathcal{T}^{c_0}\Delta c). \quad (11)$$

If $(s, a, r, s)$ is sampled from the CMDP with context $c_0$, then $\mathrm{CSE}((s, a, r, s'), c - c_0)$ is a sample $(\bar{r}, \bar{s}')$ from the approximate CMDP in context $c$. In particular, CSE allows us to sample from the approximate CMDP at perturbed contexts by performing data augmentation on samples from the original CMDP at a base context. The samples generated via this procedure might have a lower quality than those generated by exact domain randomization, due to approximation errors, but on the upside, when the derivatives are available, CSE offers a very easy to implement way to integrate structure information into the training. We provide a regularization perspective on CSE in Appendix B.6. The full algorithm for CSE in deterministic environments is shown in Algorithm 1.

## 5 Experiments

In this section, we perform numerical experiments to test CEBE and CSE. For continuous control problems, we use a simple feed-forward network and train with Soft Actor Critic (SAC) (Haarnoja et al., 2018). The inputs to the neural network are states with the context appended on (i.e., $(s, c)$). We compare against baseline training (i.e., vanilla SAC) and local domain randomization (LDR). LDR is a popular method in robotics and can provide significant generalization benefits (Jakobi et al., 1995; Levy and Wolf, 2015; Sadeghi and Levine, 2016; Tobin et al., 2017). When using LDR, one first samples a context that is a perturbation of the original training context and then produces a trajectory from the perturbed context in order to broaden the training context distribution. Since LDR has access to exact training data in a neighborhood of the original training context, we treat LDR as an idealized benchmark method for comparison.

In Section 5.1, we empirically demonstrate that the $Q$-function obtained by dynamic programming with the CEBE is first-order accurate, confirming our theory in Section 3. In Section 5.2, we

**Algorithm 1** Off-policy RL algorithm with context sample enhancement

1: Given: CMDP $\mathcal{M}$, training contexts $\mathcal{D}^{\text{train}}$, data collection iterations $N$, train iterations $M$, perturbation radius $\epsilon$, and off-policy RL algorithm ALGO.
2: Initialize policy $\pi$, value functions $Q$, and replay buffer $B$.
3: Collect some number of trajectories from a random policy in CMDP $\mathcal{M}'(c)$ with $c \sim \mathcal{D}^{\text{train}}$
4: **for** $N$ iterations **do**
5:      Sample $c \sim \mathcal{D}^{\text{train}}$
6:      Collect a trajectory $\{(s_t, c, a_t, r_t, s_{t+1})\}_{t \geq 0}$ from $\mathcal{M}'(c)$ using $\pi$ and store in buffer $B$
7:      **for** Some number of training iterations **do**
8:          Sample a batch $\{(s_t^i, c^i, a_t^i, r_t, s_{t+1}^i)\}_i$ from buffer $B$
9:          Generate perturbations $\Delta c^i \in \mathcal{B}(c^i, \epsilon)$ and compute $(\bar{r}, \bar{s}') = \text{CSE}(x^i, \Delta c^i)$
10:         Update samples $x^i \leftarrow (s, c + \Delta c^i, a, \bar{r}, \bar{s}')$ and train with ALGO on the updated batch
11:      **end for**
12: **end for**

empirically demonstrate improved performance with CSE compared to baseline training on a relatively simple continuous control environment. In Section 5.3, we test on goal-based classic control environments. In Section 5.4, we test on MuJoCo environments with context dependent tasks introduced in the work of Lee and Chung (2021). In all continuous control environments, we train 10 policies and compute the average return over 64 trajectories. We then report the mean and 95% confidence interval of the mean returns for each policy. The full list of hyperparameters and experiment configurations are included in Appendix D.

### 5.1 Tabular CEBE

We begin by examining a tabular setting where we show that the CEBE is a first-order approximation of the Bellman equation. Using dynamic programming allows us to exactly solve the Bellman equation and the CEBE and thus avoid noise due to sampling and training in deep RL. We consider the tabular Cliffwalking from gymnasium (Towers et al., 2024), in which an agent must navigate a grid-world to a goal state without first "falling" off the cliff into a terminal state. At each step, the agent has some probability $c$ of slipping into an adjacent state. To introduce nonlinearity into the reward function, we consider the following two choices:

$$R_{\text{cliff}}^c = -\frac{100}{c}, \quad R_{\text{goal}}^c = c^{-2}; \qquad R_{\text{cliff}}^c = \frac{-10}{1+c}, \quad R_{\text{goal}}^c = (1+c)^{-1.5}. \tag{12}$$

In each case, we let $R^c$ be equal to $R_{\text{cliff}}^c$ if the agent falls off the cliff, $R_{\text{goal}}^c$ if it achieves the goal state, and 0 otherwise. In Figure 1, we plot the approximation error $\|Q_{\text{CE}}^c - Q_{\text{BE}}^c\|_\infty$ on a log-log scale for 100 choices of $c$ and fit a line to the first 10 points to show that $Q_{\text{CE}}^c$ has an approximation error of $O(\|c - c_0\|^2)$. We see the best-fit line has a slope $\approx 2$ for both choices of the reward function. This numerically demonstrates Theorem 3 and shows that $Q_{\text{CE}}^c$ is a first-order approximation of $Q_{\text{BE}}^c$.

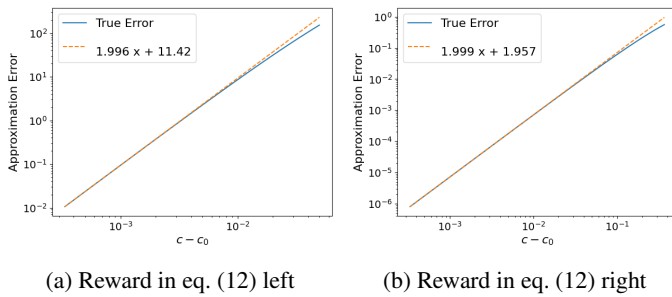

(a) Reward in eq. (12) left      (b) Reward in eq. (12) right

Figure 1: Approximation error of CEBE on Cliffwalker environment with different rewards. This experiment uses 5 rows, 6 columns, $c = 0.1$, and $\gamma = 0.9$.

## 5.2 Simple Control Environments

We begin our study of continuous control problems with environments with linear transitions and rewards. We let SimpleDirection denote the environment with transition and reward functions

$$\mathcal{T}^c(s, a) = s + a + c, \quad R^c(s, s') = s' \cdot c,$$

where $\mathcal{S} = \mathbb{R}^2$, $\mathcal{A}, \mathcal{C} = [-1, 1]^2$, and $s_0 \sim \text{Uniform}\left([-1, 1]^2\right)$. This environment was chosen because it has a reward function similar to ones we will test later in the MuJoCo environments, but has much more simple dynamics. In SimpleDirection, the agent is incentivized to move in the direction $c$ by picking action $a = \text{sign}(c)$. Figure 2 shows the evaluation results for policies trained on SimpleDirection with Baseline, CSE, and LDR as we vary the context parameters. We observe that CSE performs similarly to LDR and much better than the baseline. We provide additional analyses of this environment in Appendix E.

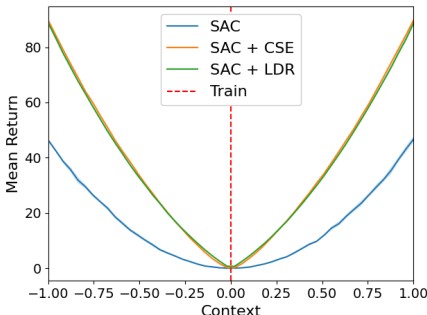

Figure 2: Comparison of training methods as we vary the first context parameter in SimpleDirection.

## 5.3 Classic Control

We consider a goal-based extension of the Pendulum environment (Towers et al., 2024), which we denote PendulumGoal. In addition to the physical parameters, we add a goal state to the context to increase the task complexity. We let $c = (g, m, l, \tau)$, where $g$ is the gravitational acceleration, $m$ the mass, $l$ the pendulum length, and $\tau \in [-1, 1]$ the desired torque at the goal state, and set

$$R^c = \pi^2 \sin\left(\frac{\theta_{\text{goal}} - \theta}{2}\right)^2 + 0.1\,\dot{\theta}^2 + 0.001\,u^2, \tag{13}$$

where $u$ is the action and $\theta_{\text{goal}} = \sin^{-1}\left(-2\tau/mgl\right)$. In Figure 3, we plot the evaluation results for each method as we vary $g$ and the goal torque $\tau$. Both CSE and LDR consistently outperform the baseline. In some contexts, CSE performs much better than LDR, e.g., when the goal torque is greater than $0.6$. We present the results of varying the other context parameters in Appendix C. We also present additional experiments with a goal-based CartPole environment (CartGoal) in Appendix C.

## 5.4 MuJoCo Environments

For this section, we use the goal-based MuJoCo environments CheetahVelocity and AntDirection introduced in the work of Lee and Chung (2021). These environments are based on the HalfCheetah and Ant environments from Todorov et al. (2012). The CheetahVelocity environment uses a modified reward function that rewards the cheetah for running at a velocity specified by the context. The AntDirection environment uses a modified reward function that rewards the ant for running in a direction specified by the context.

In Figure 4a, we show the results of the CheetahVelocity experiment. When sweeping over the goal velocity, CSE and LDR outperform the baseline for most of the contexts. CSE performs similar to LDR. When the goal velocity is between 0 and 2, CSE and LDR perform nearly as well as on the training context, while baseline starts to degrade. In CheetahVelocity, although baseline performs best when the goal velocity is greater than $2.6$, CSE still outperforms LDR. In AntDirection, shown in Figure 4b, CSE performs similarly to LDR in most contexts, though it achieves lower returns in the region $[3.5, 5]$.

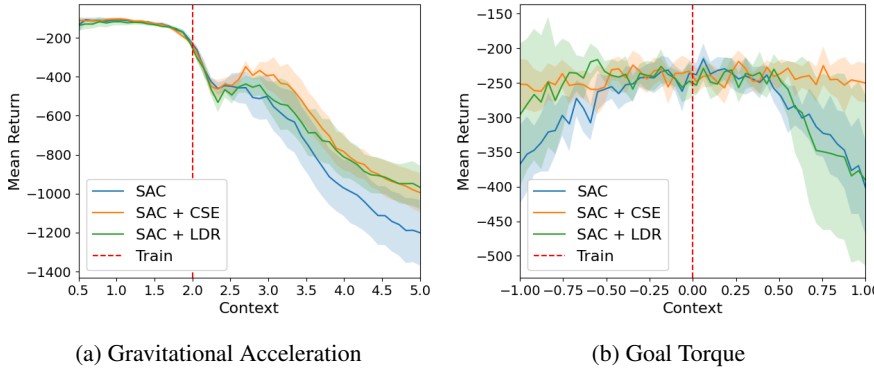

(a) Gravitational Acceleration

(b) Goal Torque

Figure 3: Comparison of training methods on PendulumGoal evaluating on different gravitational acceleration and goal torque parameters. All context parameters $(g, m, l, \tau)$ are perturbed during training.

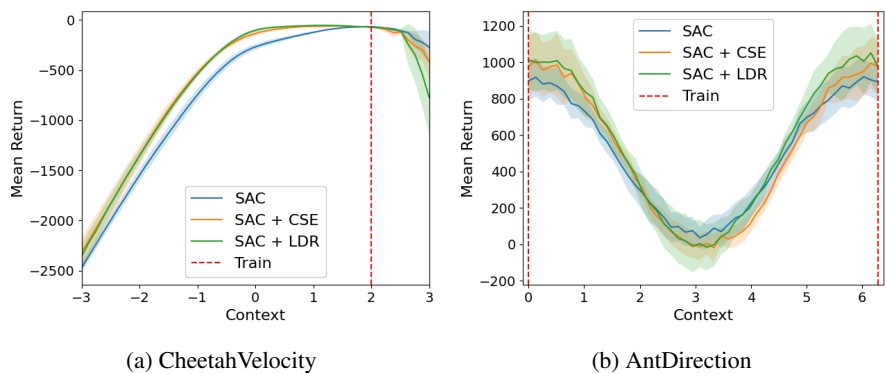

(a) CheetahVelocity

(b) AntDirection

Figure 4: Comparison of training methods on the goal-based MuJoCo environments.

# 6   Conclusion

We proposed a perturbative framework for CMDPs as an approach to improve zero-shot generalization in reinforcement learning. In particular, we introduced an approximation of the Bellman equation called the context-enhanced Bellman equation CEBE and showed theoretically that it approximates the value functions of the true CMDP in a neighborhood of contexts. Moreover, we showed that optimizing a policy based on our proposed CEBE produces a policy that is nearly optimal in the original CMDP. Using this framework we then introduced a context sample enhancement CSE procedure to generate samples that provably approximate samples from unobserved contexts. Finally, we performed experiments in diverse simulation environments. The results suggest that CSE can serve as a powerful method for improving generalization in DRL in smooth CMDPs.

We highlight that CSE is easy to implement and can be easily incorporated with other generalization methods. While we focused on model-free algorithms, CSE is compatible with model-based methods and may aid in training better world models. We think that CSE could potentially also be used to effectively sample from a larger volume of the context space in high-dimensional context spaces where domain randomization may suffer from a curse of dimensionality. However, further work is needed to understand the sample complexity of CSE in comparison to domain randomization in this setting. Another potential avenue for future study is in applying CSE to offline RL where obtaining new samples is impractical. Aside from improving generalization, one could explore different ways to leverage gradient information to design principled variations of prioritized replay buffers and exploration strategies (Jiang et al., 2021, 2023). The gradient information of the CMDP could also highlight states in the CMDP which are sensitive to context parameters and allow one to focus more training around these sensitive states or use adversarial context perturbations (Mehta et al., 2020).

We conclude by pointing limitations of our work. While our theory with CEBE applies in general to smooth CMDPs with deterministic rewards, CSE focuses on deterministic transitions which do not always occur in practice. We primarily consider this case because it provides an efficient data augmentation method. Extensions of this may consider taking gradients of appropriately defined transport maps between distributions. Another limitation is that our analysis and experiments focus on fully-observable state and context spaces. This does not always hold in practice and future study should examine the sensitivity of CSE with respect to noisy gradients in context space as well as partially observable environments.

**Acknowledgments**

This project has been supported by NSF DMS-2145630 and NSF CCF-2212520. GM also acknowledges support from DARPA AIQ in project HR00112520014, DFG SPP 2298 project 464109215, and BMFTR in DAAD project 57616814. We also thank Ruibin Lyu for his contributions during an early stage of the project, including writing test cases, verifying model components, and for helpful discussion.

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

# A   Related Works

In this section, we provide a more detailed overview of related works and commentary that could not be covered in main text.

**Contextual Multi-armed Bandits**   Multi-armed bandits (MABs) serve as a model of reinforcement learning in which the problem horizon is a single step. Contextual Multi-armed bandits (CMABs) are a natural extension of MABs which include contextual information as part of the problem specification. Contextual bandits are a popular modeling tool for a variety of applications including hyperparameter optimization and robotics (Bouneffouf et al., 2020). Lu et al. (2010) introduce a clustering algorithm for CMABs over metric spaces with sufficient regularity. Dimakopoulou et al. (2017) propose a method to improve estimation in CMABs and reduce bias for CMDPs with generalized linear reward functions. The assumption of linearity resembles our approach of approximating transition and reward functions with their linearizations.

**Challenges in CMDPs**   CMDPs present new challenges for reinforcement learning as a single CMDP can model potentially infinitely many MDPs. CMDPs can have substantial structure built into the context space that can aid in improving generalization (see the survey by Mohan et al., 2024). In our work, we do not focus specifically on one CMDP's structure, but opt for developing general methods based on first-order information of the context. Some works consider minimal assumptions on the structure of the CMDP by only imposing a notion of proximity between different MDPs in the CMDP. For example, Hallak et al. (2015) consider CMDPs with finite context spaces and introduce a clustering algorithm in context space for training across all contexts. Modi et al. (2018) establish PAC bounds on learning smooth CMDPs. These works primarily consider a zeroth-order approximation and propose covering arguments. This neglects the true changes in dynamics that occur in the CMDP, which we capture through first-order approximations of the CMDP in the context space.

Jiang et al. (2023) examine how better exploration in the training environments can improve generalization at test time. Weltevrede et al. (2024) build on this by proposing an additional exploration phase at the start of each episode to broaden the starting state distribution, which can be viewed as an implicit regularization. While these works focus on improving generalization by providing better exploration of the state space, they do not directly account for variations in the context during training.

Other works consider CMDPs where the contexts are limited during training or adversarially chosen. Ghosh et al. (2021) show that contextual MDPs can be viewed as Epistemic POMDPs when training on a limited number of contexts. Levy et al. (2023) analyze regret in CMDPs with adversarially chosen context, reflecting worst case performance as the context changes. In our work, we consider gradients to generate approximations in all context directions, including potentially adversarially chosen directions.

**RL Zero-Shot Generalization**   Benjamins et al. (2023) identify the need for context in zero-shot generalization and propose a set of environments, CARL, to test generalization in RL. Ball et al. (2021) propose model-based data-augmentation methods for offline RL, which introduce noise into the transition dynamics to reduce overfitting to the observations. This method improves generalization through data augmentation like our work, but does not directly account for changes in the context.

Cho et al. (2024) consider a related approach in which they aim to determine an optimal set of training contexts for generalization. The authors propose an acquisition function that uses a linear function to approximate the generalization gap for performance on out of distribution contexts. Learning this linear model requires access to many contexts, which could be alleviated with access to context gradients. Harrison et al. (2019) leverage linear approximations of the transition dynamics in the states and actions to improve robustness in model predictive control (MPC) when the agent has access to a simulator in the training context and is operating in a different context. Similar to our work, this method relies on a single training context.

Zhao et al. (2024) use hierarchical planning to design policies which generalize task information across contexts. They consider discrete state, action, and context spaces, evaluating primarily on gridworld environments. They design an architecture which can generalize across discrete contexts, in contrast to our focus on the training algorithm and continuous contexts.

Some works aim to learn representations of the context space. Higgins et al. (2017) use a $\beta$-VAE, and Prasanna et al. (2024) propose a contextual recurrent state space model, to learn disentangled representations of the context space. Beukman et al. (2023) introduce an architecture with a hypernetwork which uses the context to generate parameters for other parts of the network.

While several of the aforementioned works use linearity implicitly or gradient information directly, most of these works require access to multiple contexts during training. For a more comprehensive survey of zero-shot generalization in RL, we refer the reader to the survey by Kirk et al. (2023).

**Related areas** In multi-task reinforcement learning, one trains a policy to solve multiple tasks in an environment. This can improve sample complexity since knowledge of the environment can transfer across tasks. For a survey of multi-task RL, we refer the reader to the survey by Vithayathil Varghese and Mahmoud (2020). Meta-learning seeks to train a policy which can quickly adapt to new contexts. We refer the reader to the survey by Beck et al. (2025). Robust RL aims to improve generalization when model parameters are uncertain and we refer to the survey by Chen and Li (2020).

Another related direction is the simultaneous optimization of the context and policies in CMDPs. Luck et al. (2020) consider zeroth-order optimization methods for optimizing agent morphology in robotics simulations to reduce the number of contexts sampled. Thoma et al. (2024) use bilevel optimization and propose stochastic hypergradients for simultaneously optimizing high- and low-level objectives in CMDPs. While these works consider slightly different problems, they may benefit from the first-order methods we consider in our work to reduce the number of contexts sampled.

Libraries such as Procgen and C-Procgen allow for procedural environment generation (Cobbe et al., 2020; Tan et al., 2023). Not all environments support derivatives in the context variables and even those that do may not support derivatives in the context variables of interest. For a more detailed analysis of differentiable simulation, we refer the interested reader to the survey by Newbury et al. (2024).

Due to the inherent partial observability in CMDPs demonstrated by Ghosh et al. (2021), some approaches opt for using methods in POMDPs to study generalization in RL. For applications of POMDPs in robotics, see the survey of Lauri et al. (2023).

# B Theory

In this section, we prove the results stated in Section 3.

## B.1 Preliminaries

First, we introduce some notation and discuss the assumptions we make on the regularity of the MDP.

In the following, $p \in [1, \infty]$ is a constant. Recall from Section 2 that $\mathcal{A}$ refers to the space of actions available to the agent and $\mathcal{W}_p(\mathcal{A})$ denotes the space of probability measures on $\mathcal{A}$ equipped with the Wasserstein distance $W_p$.

- **Policies**: In this article, a *policy* is a Lipschitz map $\pi : \mathcal{S} \to \mathcal{W}_p(\mathcal{A})$ such that for all $s \in \mathcal{S}$, $\pi(s)$ is a probability measure. For a policy $\pi$, we will often write $\pi_s$ in place of $\pi(s)$.

- **$Q$-functions**: A *$Q$-function* is an element of $\mathrm{Lip}(\mathcal{S} \times \mathcal{A})$. We will often write $Q_s$ to denote the function $a \mapsto Q(s, a)$.

- **Reward functions**: We assume that the reward function $R$ of the MDP is an element of $\mathrm{Lip}(\mathcal{S} \times \mathcal{A})$.

- **Transition maps**: A *transition map* is a Lipschitz mapping $\mathcal{T} : \mathcal{S} \times \mathcal{A} \to \mathcal{W}_p(\mathcal{S})$. For a transition map $\mathcal{T}$, we will often write $\mathcal{T}_{s,a}$ in place of $\mathcal{T}(s, a)$.

- **Bellman operator**: If $\mathcal{T} : \mathcal{S} \times \mathcal{A} \to \mathcal{W}_p(\mathcal{S})$ is a transition map and $\pi : \mathcal{S} \to \mathcal{W}_p(\mathcal{A})$ is a policy, we define the operator $A^{\mathcal{T}, \pi} : \mathrm{Lip}(\mathcal{S} \times \mathcal{A}) \to \mathrm{Lip}(\mathcal{S} \times \mathcal{A})$ by

$$(A^{\mathcal{T}, \pi} Q)(s, a) := \int_{\mathcal{S}} \int_{\mathcal{A}} Q_{s'}(a') d\pi_{s'}(a') d\mathcal{T}_{s,a}(s'). \tag{14}$$

We prove that this operator is well-defined in Lemma 7. Observe that the Bellman equation can be written as $Q = R + \gamma A^{\mathcal{T}, \pi} Q$, where $\gamma \in (0, 1)$ is the discount factor.

## B.2 Proof of Theorem 1

We need to show that the operator $A^{\mathcal{T},\pi}$ is well-defined, in the sense that it maps elements of $\text{Lip}(\mathcal{S} \times \mathcal{A})$ to elements of $\text{Lip}(\mathcal{S} \times \mathcal{A})$, and that it is a bounded operator on $\text{Lip}(\mathcal{S} \times \mathcal{A})$. To this end, we prove a couple of lemmas establishing the regularity of Lipschitz functions under integration.

**Lemma 5.** *Let $X$ be a metric space. Let $f \in \text{Lip}(X)$ and let $\nu_1, \nu_2 \in \mathcal{W}_p(X)$. Then*

$$\left| \int_X f d\nu_1 - \int_X f d\nu_2 \right| \leq L_f W_p(\nu_1, \nu_2).$$

*Proof.* First we prove the statement for the case $p < \infty$. Let $\epsilon > 0$ and let $p'$ be the Hölder conjugate of $p$. There exists a coupling $\Gamma \in \Pi(\nu_1, \nu_2)$ with

$$\left( \int_{X \times X} d(x,y)^p d\Gamma(x,y) \right)^{1/p} < W_p(\nu_1, \nu_2) + \epsilon.$$

Then

$$\left| \int_X f d\nu_1 - \int_X f d\nu_2 \right| = \left| \int_X \int_X f(x) d\Gamma(x,y) - \int_X \int_X f(y) d\Gamma(x,y) \right|$$

$$\leq \int_X \int_X |f(x) - f(y)| d\Gamma(x,y)$$

$$\leq \left( \int_X \int_X |f(x) - f(y)|^p d\Gamma(x,y) \right)^{1/p} \left( \int_X \int_X 1^{p'} d\Gamma(x,y) \right)^{1/p'}$$

$$\leq \left( \int_X \int_X L_f^p d(x,y)^p d\Gamma(x,y) \right)^{1/p}$$

$$\leq L_f(W_p(\nu_1, \nu_2) + \epsilon),$$

where we applied Hölder's inequality in the third line. Since this holds for all $\epsilon > 0$, we have

$$\left| \int_X f d\nu_1 - \int_X f d\nu_2 \right| \leq L_f W_p(\nu_1, \nu_2).$$

For the case $p = \infty$, simply take the limit of the above inequality as $p \to \infty$. $\qquad\square$

**Lemma 6.** *Let $X$ and $Y$ be metric spaces. Let $f \in \text{Lip}(X \times Y)$, and let $\mu : X \to \mathcal{W}_p(Y)$ be Lipschitz with constant $L_\mu$. We write $\mu_x$ in place of $\mu(x)$. Then the function $g : X \to \mathbb{R}$ defined by*

$$g(x) := \int_Y f_x d\mu_x$$

*is in $\text{Lip}(X)$. Moreover,*

$$\sup_{x \in X} |g(x)| \leq \sup_{(x,y) \in X \times Y} |f(x,y)|$$

*and*

$$L_g \leq L_f(1 + L_\mu).$$

*Proof.* By the triangle inequality,

$$\sup_{x \in X} |g(x)| \leq \sup_{x \in X} \int_Y |f_x| d\mu_x$$

$$\leq \sup_{x \in X} \sup_{y \in Y} |f(x,y)|.$$

Let $x_1, x_2 \in X$ be distinct points. Then

$$|g(x_1) - g(x_2)| = \left| \int_Y f_{x_1} d\mu_{x_1} - \int_Y f_{x_2} d\mu_{x_2} \right|$$

$$\leq \left| \int_Y f_{x_1} d\mu_{x_1} - \int_Y f_{x_2} d\mu_{x_1} \right| + \left| \int_Y f_{x_2} d\mu_{x_1} - \int_Y f_{x_2} d\mu_{x_2} \right|.$$

We bound the terms of the above inequality separately. For the first term,

$$\left| \int_Y f_{x_1} d\mu_{x_1} - \int_Y f_{x_2} d\mu_{x_1} \right| \leq \int_Y |f_{x_1} - f_{x_2}| d\mu_{x_1}$$

$$\leq \int_Y L_f d(x_1, x_2) d\mu_{x_1}$$

$$= L_f d(x_1, x_2).$$

For the second term, by Lemma 5 we have

$$\left| \int_Y f_{x_2} d\mu_{x_1} - \int_Y f_{x_2} d\mu_{x_2} \right| \leq L_f W_p(\mu_{x_1}, \mu_{x_2})$$

$$\leq L_f L_\mu d(x_1, x_2).$$

Combining the two terms, we get

$$|g(x_1) - g(x_2)| \leq L_f d(x_1, x_2) + L_f L_\mu d(x_1, x_2),$$

so $g$ has Lipschitz constant $L_f(1 + L_\mu)$. $\qquad\square$

**Lemma 7** ($A^{\mathcal{T},\pi}$ is well-defined)**.** *The linear map $A^{\mathcal{T},\pi} : \mathrm{Lip}(\mathcal{S} \times \mathcal{A}) \to \mathrm{Lip}(\mathcal{S} \times \mathcal{A})$ given in (14) is well-defined. Moreover, for all $Q \in \mathrm{Lip}(\mathcal{S} \times \mathcal{A})$,*

$$\|A^{\mathcal{T},\pi} Q\|_\infty \leq \|Q\|_\infty$$

*and*

$$L_{A^{\mathcal{T},\pi} Q} \leq L_Q L_{\mathcal{T}} (1 + L_\pi).$$

*Proof.* Let $Q \in \mathrm{Lip}(\mathcal{S} \times \mathcal{A})$. First, observe that

$$\sup_{(s,a) \in \mathcal{S} \times \mathcal{A}} |(A^{\mathcal{T},\pi} Q)(s,a)| \leq \sup_{(s,a) \in \mathcal{S} \times \mathcal{A}} \int_{\mathcal{S}} \int_{\mathcal{A}} |Q_{s'}(a')| d\pi_{s'}(a') dT_{s,a}(s')$$

$$\leq \sup_{(s',a') \in \mathcal{S} \times \mathcal{A}} |Q(s', a')|,$$

so $\|A^{\mathcal{T},\pi} Q\|_\infty \leq \|Q\|_\infty$.

Next, let $\varphi : \mathcal{S} \to \mathbb{R}$ be defined by

$$\varphi(s) := \int_{\mathcal{A}} Q_s d\pi_s.$$

By Lemma 6, $\varphi \in \mathrm{Lip}(\mathcal{S})$ and

$$L_\varphi \leq L_Q(1 + L_\pi).$$

Now

$$(A^{\mathcal{T},\pi} Q) = \int_{\mathcal{S}} \varphi(s') d\mathcal{T}_{s,a}(s').$$

By Lemma 5,

$$L_{A^{\mathcal{T},\pi} Q} \leq L_\varphi L_{\mathcal{T}}$$

$$\leq L_Q L_{\mathcal{T}} (1 + L_\pi).$$

So $A^{\mathcal{T},\pi}$ maps elements of $\mathrm{Lip}(\mathcal{S})$ to $\mathrm{Lip}(\mathcal{S})$, and

$$\|A^{\mathcal{T},\pi}\| \leq \max(1, L_{\mathcal{T}}(1 + L_\pi)).$$

$\qquad\square$

Next, we show that the regularity of the transition and reward functions implies regularity of the solution to the Bellman equation.

**Lemma 8** ($Q$ is uniquely defined and Lipschitz)**.** *Let $\gamma < \max\left(1, \frac{1}{L_{\mathcal{T}}(1+L_\pi)}\right)$ and suppose that $R \in \mathrm{Lip}(\mathcal{S} \times \mathcal{A})$. Then there exists a unique solution $Q \in \mathrm{Lip}(\mathcal{S} \times \mathcal{A})$ to the Bellman equation*

$$Q = R + \gamma A^{\mathcal{T},\pi} Q. \tag{15}$$

*Proof.* Consider the power series

$$B := \sum_{n=0}^{\infty} \gamma^n (A^{\mathcal{T},\pi})^n.$$

By Lemma 7, $\|A^{\mathcal{T},\pi}\| \le \max(1, L_{\mathcal{T}}(1+L_\pi))$. Then we have $\|\gamma A^{\mathcal{T},\pi}\| < 1$, so this power series converges absolutely in the operator norm. Then

$$\begin{aligned} B(I - \gamma A^{\mathcal{T},\pi}) &= \sum_{n=0}^{\infty} \gamma^n (A^{\mathcal{T},\pi})^n - \sum_{n=0}^{\infty} \gamma^{n+1} (A^{\mathcal{T},\pi})^{n+1} \\ &= I. \end{aligned}$$

The same calculation shows that $(I - \gamma A^{\mathcal{T},\pi})B = I$, so $B = (I - \gamma A^{\mathcal{T},\pi})^{-1}$. Now $Q$ satisfies the Bellman equation if and only if

$$(I - \gamma A^{\mathcal{T},\pi})Q = R.$$

Since $I - \gamma A^{\mathcal{T},\pi}$ is invertible as an operator on $\mathrm{Lip}(\mathcal{S} \times \mathcal{A})$, there exists a unique $Q \in \mathrm{Lip}(\mathcal{S} \times \mathcal{A})$ satisfying this equation. $\qquad\square$

**Lemma 9** ($\mathcal{T}$-stability of $A^{\mathcal{T},\pi}$)**.** *Let $\mathcal{T}^{(1)}, \mathcal{T}^{(2)} : \mathcal{S} \times \mathcal{A} \to \mathcal{W}_p(\mathcal{S})$ be transition functions with*

$$\sup_{(s,a)\in\mathcal{S}\times\mathcal{A}} W_p(\mathcal{T}^{(1)}(s,a), \mathcal{T}^{(2)}(s,a)) \le \delta.$$

*Then*

$$\|A^{\mathcal{T}^{(1)},\pi} - A^{\mathcal{T}^{(2)},\pi}\| \le (1 + L_\pi)\delta.$$

*Proof.* As in the proof of Lemma 7, let $Q \in \mathrm{Lip}(\mathcal{S} \times \mathcal{A})$, and let $\varphi : \mathcal{S} \to \mathbb{R}$ be defined by

$$\varphi(s) := \int_{\mathcal{A}} Q_s d\pi_s.$$

By Lemma 6, $\varphi \in \mathrm{Lip}(\mathcal{S})$ and

$$L_\varphi \le L_Q(1 + L_\pi) \le \|Q\|(1 + L_\pi).$$

Now by Lemma 5

$$\begin{aligned} \left|(A^{\mathcal{T}^{(1)},\pi}Q)(s,a) - (A^{\mathcal{T}^{(2)},\pi}Q)(s,a)\right| &= \left|\int_{\mathcal{S}} \varphi(s')d\mathcal{T}_{s,a}^{(1)} - \int_{\mathcal{S}} \varphi(s')d\mathcal{T}_{s,a}^{(2)}\right| \\ &\le L_\varphi W_p(\mathcal{T}^{(1)}(s,a), \mathcal{T}^{(2)}(s,a)) \\ &\le \|Q\|(1 + L_\pi)\delta. \end{aligned}$$

The result follows by taking a supremum over $(s,a) \in \mathcal{S} \times \mathcal{A}$ and $Q \in \mathrm{Lip}(\mathcal{S} \times \mathcal{A})$. $\qquad\square$

**Proof of Theorem 1**

*Proof.* Let $A_1 = A^{\mathcal{T}^{(1)},\pi}$ and let $A_2 = A^{\mathcal{T}^{(2)},\pi}$. By Lemma 9,

$$\|A_1 - A_2\| \le (1 + L_\pi)\delta_T.$$

Since $Q^{(1)}$ and $Q^{(2)}$ are solutions to Bellman equations, we have

$$\begin{aligned} Q^{(1)} - Q^{(2)} &= (R^{(1)} + \gamma A_1 Q^{(1)}) - (R^{(2)} + \gamma A_2 Q^{(2)}) \\ &= (R^{(1)} - R^{(2)}) + (\gamma A_1 Q^{(1)} - \gamma A_1 Q^{(2)}) + (\gamma A_1 Q^{(2)} - \gamma A_2 Q^{(2)}). \end{aligned}$$

Then by the triangle inequality,
$$\|Q^{(1)} - Q^{(2)}\|_\infty \le \|R^{(1)} - R^{(2)}\|_\infty + \gamma\|A_1 Q^{(1)} - A_1 Q^{(2)}\|_\infty + \gamma\|A_1 Q^{(2)} - A_2 Q^{(2)}\|_\infty.$$
$$\le \|R^{(1)} - R^{(2)}\|_\infty + \gamma\|A_1 Q^{(1)} - A_1 Q^{(2)}\|_\infty + \gamma\|A_1 - A_2\|\|Q^{(2)}\|_{\mathrm{Lip}(\mathcal{S}\times\mathcal{A})}$$
$$\le \|R^{(1)} - R^{(2)}\|_\infty + \gamma\|A_1 Q^{(1)} - A_1 Q^{(2)}\|_\infty + \gamma(1 + L_\pi)\delta_T\|Q^{(2)}\|_{\mathrm{Lip}(\mathcal{S}\times\mathcal{A})}.$$

By assumption, we have $\|R^{(1)} - R^{(2)}\|_\infty \le \delta_R$. By Lemma 7, we have
$$\|A_1 Q^{(1)} - A_1 Q^{(2)}\|_\infty \le \|Q^{(1)} - Q^{(2)}\|_\infty.$$

By the Bellman equation for $Q^{(2)}$,
$$\|Q^{(2)}\|_{\mathrm{Lip}(\mathcal{S}\times\mathcal{A})} = \|(I - \gamma A_2)^{-1} R^{(2)}\|$$
$$\le \|(I - \gamma A_2)^{-1}\|\|R^{(2)}\|_{\mathrm{Lip}(\mathcal{S}\times\mathcal{A})}$$
$$\le (1 - \gamma\|A_2\|)^{-1}\|R^{(2)}\|_{\mathrm{Lip}(\mathcal{S}\times\mathcal{A})}.$$

By Lemma 7,
$$\|A_2\| \le \max(1, L_{\mathcal{T}^{(2)}}(1 + L_\pi)),$$

so
$$\|Q^{(2)}\|_{\mathrm{Lip}(\mathcal{S}\times\mathcal{A})} \le (1 - \gamma\|A_2\|)^{-1}\|R^{(2)}\|_{\mathrm{Lip}(\mathcal{S}\times\mathcal{A})}$$
$$\le \frac{\|R^{(2)}\|_{\mathrm{Lip}(\mathcal{S}\times\mathcal{A})}}{1 - \gamma\max(1, L_{\mathcal{T}^{(2)}}(1 + L_\pi))}.$$

Combining, we get
$$\|Q^{(1)} - Q^{(2)}\|_\infty \le \delta_R + \gamma\|Q^{(1)} - Q^{(2)}\|_\infty + \frac{\gamma(1 + L_\pi)\delta_T\|R^{(2)}\|_{\mathrm{Lip}(\mathcal{S}\times\mathcal{A})}}{1 - \gamma\max(1, L_{\mathcal{T}^{(2)}}(1 + L_\pi))}$$

and thus
$$\|Q^{(1)} - Q^{(2)}\|_\infty \le \frac{1}{1 - \gamma}\left(\delta_R + \frac{\gamma(1 + L_\pi)\delta_T\|R^{(2)}\|_{\mathrm{Lip}(\mathcal{S}\times\mathcal{A})}}{1 - \gamma\max(1, L_{\mathcal{T}^{(2)}}(1 + L_\pi))}\right).$$

□

## B.3 Deterministic CEBE approximation error

In this section we prove Theorem 2. We first reiterate some of the assumptions implicit in the statement of the theorem and stated before.

- The state space $\mathcal{S}$ and the action space $\mathcal{A}$ are Banach spaces.
- The deterministic transition function $\mathcal{T} : \mathcal{S} \times \mathcal{A} \times \mathcal{C} \to \mathcal{W}_p(\mathcal{S})$ is of the form $\mathcal{T}(s, a) = \delta_{f(s,a,c)}$, where $f : \mathcal{S} \times \mathcal{A} \times \mathcal{C} \to \mathcal{S}$ is twice continuously differentiable with bounded second partial derivatives. We will write $f^c$ to denote the function $f(\bullet, \bullet, c)$.
- The reward function $R : \mathcal{S} \times \mathcal{A} \times \mathcal{C} \to \mathbb{R}$ is twice continuously differentiable with bounded second partial derivatives. We will write $R^c$ to denote the function $R(\bullet, \bullet, c)$.
- The policy $\pi : \mathcal{S} \times \mathcal{C} \to \mathcal{W}_p(\mathcal{A})$ is Lipschitz.

**Lemma 10.** *Let $X$ be a metric space. Then for all $x_1, x_2 \in X$ and $p \in [1, \infty]$, $W_p(\delta_{x_1}, \delta_{x_2}) \le d(x_1, x_2)$.*

*Proof.* First we prove the statement for $p < \infty$. Let $\Gamma$ be the coupling between $\delta_{x_1}$ and $\delta_{x_2}$ given by the product measure. Then

$$W_p(\delta_{x_1}, \delta_{x_2}) \le \left(\int_X \int_X d(y_1, y_2)^p d\Gamma(y_1, y_2)\right)^{1/p}$$
$$\le \left(\int_X \int_X d(y_1, y_2)^p d(\delta_{x_1}(y_1)\delta_{x_2}(y_2))\right)^{1/p}$$
$$= (d(x_1, x_2)^p)^{1/p}$$
$$= d(x_1, x_2).$$

For the case $p = \infty$, take the limit as $p \to \infty$ in the above bound. □

*Proof of Theorem 2.* In view of Theorem 1, we need to show that the linearized transition and reward functions are good estimates of their true values. Let us define $f_{\mathrm{CE}} : \mathcal{S} \times \mathcal{A} \times \mathcal{C} \to \mathcal{S}$ by

$$f_{\mathrm{CE}}^c(s, a) = f^{c_0}(s, a) + \partial_c f^{c_0}(s, a) \cdot (c - c_0),$$

so that $\mathcal{T}_{\mathrm{CE}}^c(s, a) = \delta_{f_{\mathrm{CE}}^c(s,a)}$.

Let $s \in \mathcal{S}$, $a \in \mathcal{A}$, and $c \in \mathcal{C}$. By Taylor's theorem, there exists $c' \in [c_0, c]$ such that

$$\|f^c(s, a) - f^{c_0}(s, a) - \partial_c f^{c_0}(s, a) \cdot (c - c_0)\| \leq \tfrac{1}{2}\|\partial_c^2 f^{c'}(s, a)\|\|c - c_0\|^2$$

which we can write as

$$\|f^c(s, a) - f_{\mathrm{CE}}^c(s, a)\| \leq \|D^2 f\|_\infty \|c - c_0\|^2.$$

Here $\partial_c$ denotes the partial derivative with respect to $c$ while $D$ denotes the derivative with respect to all parameters. By Lemma 10,

$$W_p(\mathcal{T}^c(s, a), \mathcal{T}_{\mathrm{CE}}^c(s, a)) \leq \|f^c(s, a) - f_{\mathrm{CE}}^c(s, a)\|$$
$$\leq \|D^2 f\|_\infty \|c - c_0\|^2.$$

Hence we have shown that the transition maps are close.

Next we show that the reward functions are close. Again by Taylor's theorem, there exists $c' \in [c_0, c]$ such that

$$|R^c(s, a) - R^{c_0}(s, a) - \partial_c R^{c_0}(s, a) \cdot (c - c_0)| \leq \|D^2 R^{c'}(s, a)\|\|c - c_0\|^2$$
$$\leq \|D^2 R\|_\infty \|c - c_0\|^2.$$

So

$$\|R^c - R_{\mathrm{CE}}^c\|_\infty \leq \|D^2 R\|_\infty \|c - c_0\|^2,$$

and the reward functions are close.

Next we will show that the transition and reward functions are Lipschitz. By Lemma 10, for all $(s_1, a_1), (s_2, a_2) \in \mathcal{S} \times \mathcal{A}$, we have

$$W_p(\delta_{f^c(s_1,a_1)}, f^c(s_2,a_2)) \leq \|f^c(s_1, a_1) - f^c(s_2, a_2)\|$$
$$\leq \|\partial_{(s,a)} f^c\|_\infty \|(s_1 - s_2, a_1 - a_2)\|$$

so $L_{\mathcal{T}^c} \leq \|Df\|_\infty$. Similarly,

$$W_p(\mathcal{T}_{\mathrm{CE}}^c(s_1, a_1), \mathcal{T}_{\mathrm{CE}}^c(s_2, a_2))$$
$$\leq \|f_{\mathrm{CE}}^c(s_1, a_1) - f_{\mathrm{CE}}^c(s_2, a_2)\|$$
$$\leq \|\partial_{(s,a)} f_{\mathrm{CE}}^c\|_\infty \|(s_1 - s_2, a_1 - a_2)\|$$
$$\leq \|\partial_{(s,a)} f^{c_0}(s, a) + \partial_{(s,a)} \partial_c f^{c_0}(s, a) \cdot (c - c_0)\|_\infty \|(s_1 - s_2, a_1 - a_2)\|$$
$$\leq \|Df\|_\infty \|(s_1 - s_2, a_1 - a_2)\| + \|D^2 f\|_\infty \|c - c_0\|\|(s_1 - s_2, a_1 - a_2)\|,$$

so $L_{\mathcal{T}_{\mathrm{CE}}^c} \leq \|Df\|_\infty + \|D^2 f\|_\infty \|c - c_0\|$.

Since $R$ is twice continuously differentiable, it is Lipschitz with constant $\|DR\|_\infty$, and therefore

$$\|R^c\|_{\mathrm{Lip}(\mathcal{S} \times \mathcal{A})} \leq \max(\|R\|_\infty, \|DR\|_\infty).$$

For the context-enhanced reward function,

$$\|R_{\mathrm{CE}}^c\|_{\mathrm{Lip}(\mathcal{S} \times \mathcal{A})} = \|R^{c_0} + \partial_c R^{c_0} \cdot (c - c_0)\|_{\mathrm{Lip}(\mathcal{S} \times \mathcal{A})}$$
$$\leq \|R^{c_0}\|_{\mathrm{Lip}(\mathcal{S} \times \mathcal{A})} + \|\partial_c R^{c_0} \cdot (c - c_0)\|_{\mathrm{Lip}(\mathcal{S} \times \mathcal{A})}$$
$$\leq \max(\|R\|_\infty, \|DR\|_\infty) + \|c - c_0\| \max(\|DR\|_\infty, \|D^2 R\|_\infty).$$

Now we can apply Theorem 1. Note that

$$\gamma < \frac{1}{(\|Df\|_\infty + \|D^2 f\|_\infty \|c - c_0\|)(1 + L_\pi)}$$
$$\leq \frac{1}{\max(L_{\mathcal{T}^c}, L_{\mathcal{T}_{\mathrm{CE}}^c})(1 + L_\pi)},$$

so

$$\|Q_{\mathrm{CE}}^c - Q^c\|_\infty \leq \frac{\|c - c_0\|^2}{1 - \gamma}\left(\|D^2 R\|_\infty + \frac{\gamma(1 + L_\pi)\|D^2 f\|_\infty \|R\|_{\mathrm{Lip}(\mathcal{S} \times \mathcal{A} \times \mathcal{C})}}{1 - \gamma \max(1, L_{f^c}(1 + L_\pi))}\right)$$

$$\leq \frac{\|c - c_0\|^2}{1 - \gamma}\left(\|D^2 R\|_\infty + \frac{\gamma(1 + L_\pi)\|D^2 f\|_\infty(\|R\|_\infty + \|DR\|_\infty)}{1 - \gamma \max(1, \|Df\|_\infty(1 + L_\pi))}\right).$$

$\square$

## B.4 Stochastic CEBE approximation error

In this section we provide a proof of Theorem 3. We start by introducing assumptions and notation.

- The state space $\mathcal{S}$ is a bounded metric space. The action space $\mathcal{A}$ is a metric space.

- For a metric space $X$, let $\mathrm{Meas}(X)$ denote the space of finite signed Borel measures on $X$ equipped with the total variation norm. The transition map is a function $\mathcal{T} : \mathcal{S} \times \mathcal{A} \times \mathcal{C} \to \mathrm{Meas}(\mathcal{S})$, and we suppose it is twice continuously differentiable in $\mathcal{C}$, and $\|\mathcal{T}\|_\infty, \|\partial_c \mathcal{T}\|_\infty, \|\partial_c^2 \mathcal{T}\|_\infty < \infty$. We also assume that $\mathcal{T}$ and $\partial_c \mathcal{T}$ are Lipschitz. In this subsection, we view $\mathcal{T}$ as a map into $\mathrm{Meas}(\mathcal{S})$ rather than into $\mathcal{W}_p(\mathcal{S})$. So, for example, the Lipschitz constant $L_\mathcal{T}$ refers to the Lipschitz constant of $\mathcal{T}$ as a map $\mathcal{S} \times \mathcal{A} \times \mathcal{C} \to \mathrm{Meas}(\mathcal{S})$.

- Recall that if $X$ is a metric space and $\mu \in \mathrm{Meas}(X)$, then $\mu$ admits a unique Hahn-Jordan decomposition $\mu = \mu^+ - \mu^-$. Let $\mathcal{U}(X)$ denote the open subset of $\mathrm{Meas}(X)$ consisting of the signed measures $\mu$ with $\mu^+(X) > 0$, and let $\Delta(X)$ denote the probability simplex of $\mathrm{Meas}(X)$, consisting of the (unsigned) measures $\mu$ with $\mu(X) = 1$. We define the map $P : \mathcal{U}(X) \to \Delta(X)$ by $P(\mu) := \frac{\mu^+}{\|\mu^+\|}$.

- We suppose the reward function $R : \mathcal{S} \times \mathcal{A} \times \mathcal{C} \to \mathbb{R}$ is twice continuously differentiable with respect to $\mathcal{C}$, and $\|R\|_\infty, \|\partial_c R\|_\infty, \|\partial_c^2 R\|_\infty < \infty$. Moreover, we assume that $R$ is Lipschitz.

A technical difficulty with transition maps in this setting is that the linearization of the parametrization map might produce measures that have negative components and therefore might no longer be probability measures. To fix this issue, we project the linearized transition maps onto the probability simplex (note this is not necessarily on a finite sample space). But first, we must check that it is possible to make this projection. In particular, for a projection $P$ of the form $\mathcal{U}(\mathcal{S}) \to \Delta(\mathcal{S})$ we need to check that the linearization $\mathcal{T}_{\mathrm{CE}}^c(s, a) = \mathcal{T}^{c_0}(s, a) + \partial_c \mathcal{T}^{c_0}(s, a) \cdot (c - c_0)$ lies in $\mathcal{U}(\mathcal{S})$.

**Lemma 11.** *Let $X$ be a metric space, and let $\mu \in \mathcal{U}(X)$ be such that $\mu(X) = 1$. Then*

$$\|\mu - P(\mu)\| \leq 2\|\mu^-\|.$$

*Proof.* By the triangle inequality,

$$\|\mu - P(\mu)\| \leq \|\mu - \mu^+\| + \|\mu^+ - P(\mu)\|$$

$$= \|\mu^-\| + \left\|\mu^+ - \frac{\mu^+}{\|\mu^+\|}\right\|$$

$$= \|\mu^-\| + \|\mu^+\|\left(1 - \frac{1}{\|\mu^+\|}\right)$$

$$= \|\mu^+\| + \|\mu^-\| - 1$$

$$= (1 + \|\mu^-\|) + \|\mu^-\| - 1$$

$$= 2\|\mu^-\|.$$

In the second to last line, we used that $\mu(X) = 1$, and therefore that $\|\mu^+\| - \|\mu^-\| = 1$. $\square$

**Lemma 12.** *For all $s \in \mathcal{S}$, $a \in \mathcal{A}$, and $c, c_0 \in \mathcal{C}$, we have*

$$\left(\mathcal{T}^{c_0}(s, a) + \partial_c \mathcal{T}^{c_0}(s, a) \cdot (c - c_0)\right)(\mathcal{S}) = 1.$$

*Proof.* Let $\epsilon > 0$ be such that a ball of radius $\epsilon \|c - c_0\|$ about $c_0$ is contained in $\mathcal{C}$. Let us define $\gamma : (-\epsilon, \epsilon) \to \text{Meas}(\mathcal{S})$ by

$$\gamma(t) := \mathcal{T}^{c_0 + t(c - c_0)}(s, a).$$

Since $\mathcal{T}$ is twice continuously differentiable in $\mathcal{C}$, $\gamma$ is twice continuously differentiable. Since $\gamma(t) \in \text{Meas}(\mathcal{S})$ for all $t$, we have

$$\int_{\mathcal{S}} \gamma(t) = 1.$$

So for all $t \neq 0$,

$$\int_{\mathcal{S}} \frac{\gamma(t) - \gamma(0)}{t} = 0.$$

For each $n \in \mathbb{N}$, let $\mu_n := \frac{\gamma(1/n) - \gamma(0)}{1/n}$. Since $\gamma$ is differentiable, we have $\mu_n \to \dot{\gamma}(0)$ in $\text{Meas}(\mathcal{S})$. So

$$\int_{\mathcal{S}} \dot{\gamma}(0) = \int_{\mathcal{S}} \lim_{n \to \infty} \mu_n$$
$$= \lim_{n \to \infty} \int_{\mathcal{S}} \mu_n$$
$$= 0.$$

Finally, by the chain rule, we have

$$(\mathcal{T}^{c_0}(s, a) + \partial_c \mathcal{T}^{c_0}(s, a) \cdot (c - c_0))(\mathcal{S}) = (\mathcal{T}^{c_0}(s, a) + \dot{\gamma}(0))(\mathcal{S}) = 1.$$

$\square$

The following lemma shows that the linearization $\mathcal{T}^{c_0}(s, a) + \partial_c \mathcal{T}^{c_0}(s, a) \cdot (c - c_0)$ lies in $\mathcal{U}(\mathcal{S})$, and therefore that the projection $P(\mathcal{T}^{c_0}(s, a) + \partial_c \mathcal{T}^{c_0}(s, a) \cdot (c - c_0))$ is well-defined.

**Lemma 13.** *Let $s \in \mathcal{S}$, $a \in \mathcal{A}$, and $c, c_0 \in \mathcal{C}$. If $\|c - c_0\| < (2\|\partial_c^2 \mathcal{T}\|_\infty)^{-1/2}$, then the following conditions hold:*

$$\left\| (\mathcal{T}^{c_0}(s, a) + \partial_c \mathcal{T}^{c_0}(s, a) \cdot (c - c_0))^+ \right\| \geq 1 - \|\partial_c^2 \mathcal{T}\|_\infty \|c - c_0\|^2, \quad (16a)$$

$$\mathcal{T}^{c_0}(s, a) + \partial_c \mathcal{T}^{c_0}(s, a) \cdot (c - c_0) \in \mathcal{U}(\mathcal{S}), \quad (16b)$$

$$\|\mathcal{T}^c(s, a) - P(\mathcal{T}^{c_0}(s, a) + \partial_c \mathcal{T}^{c_0}(s, a) \cdot (c - c_0))\| \leq 3\|\partial_c^2 \mathcal{T}\|_\infty \|c - c_0\|^2. \quad (16c)$$

*Proof.* By Taylor's theorem,

$$\|\mathcal{T}^c(s, a) - \mathcal{T}^{c_0}(s, a) - \partial_c \mathcal{T}^{c_0}(s, a) \cdot (c - c_0)\| \leq \|\partial_c^2 \mathcal{T}\|_\infty \|c - c_0\|^2.$$

Then by the triangle inequality,

$$\left\| (\mathcal{T}^{c_0}(s, a) + \partial_c \mathcal{T}^{c_0}(s, a) \cdot (c - c_0))^+ \right\|$$
$$\geq \left\| (\mathcal{T}^c(s, a))^+ \right\| - \|\mathcal{T}^c(s, a) - \mathcal{T}^{c_0}(s, a) - \partial_c \mathcal{T}^{c_0}(s, a) \cdot (c - c_0)\|$$
$$\geq \left\| (\mathcal{T}^c(s, a))^+ \right\| - \|\partial_c^2 \mathcal{T}\|_\infty \|c - c_0\|^2$$
$$= 1 - \|\partial_c^2 \mathcal{T}^c(s, a)\|_\infty \|c - c_0\|^2.$$

Here we used that $\mathcal{T}^c(s, a)$ is a probability measure on the second line. This establishes (16a). In particular, the last line is greater than 0 by the assumption on the norm of $\|c - c_0\|$, so

$$\left\| (\mathcal{T}^{c_0}(s, a) + \partial_c \mathcal{T}^{c_0}(s, a) \cdot (c - c_0))^+ \right\| > 0,$$

which establishes (16b).

By Lemma 12, we have $(\mathcal{T}^{c_0}(s, a) + \partial_c \mathcal{T}^{c_0}(s, a))(\mathcal{S}) = 1$. We can also bound the negative part of the linearized measure:

$$\left\| \left( \mathcal{T}^{c_0}(s, a) + \partial_c \mathcal{T}^{c_0}(s, a) \cdot (c - c_0) \right)^- \right\|$$
$$\leq \left\| (\mathcal{T}^c(s, a))^- \right\| + \left\| \mathcal{T}^c(s, a) - \mathcal{T}^{c_0}(s, a) - \partial_c \mathcal{T}^{c_0}(s, a) \cdot (c - c_0) \right\|$$
$$\leq \left\| (\mathcal{T}^c(s, a))^- \right\| + \| \partial_c^2 \mathcal{T} \|_\infty \| c - c_0 \|^2$$
$$= \| \partial_c^2 \mathcal{T} \|_\infty \| c - c_0 \|^2.$$

Then by Lemma 11,

$$\| \mathcal{T}^{c_0}(s, a) + \partial_c \mathcal{T}^{c_0}(s, a) \cdot (c - c_0) - P(\mathcal{T}^{c_0}(s, a) + \partial_c \mathcal{T}^{c_0}(s, a) \cdot (c - c_0)) \|$$
$$\leq 2 \left\| \left( \mathcal{T}^{c_0}(s, a) + \partial_c \mathcal{T}^{c_0}(s, a) \cdot (c - c_0) \right)^- \right\|$$
$$\leq 2 \| \partial_c^2 \mathcal{T} \|_\infty \| c - c_0 \|^2.$$

To conclude, we apply the triangle inequality:

$$\| \mathcal{T}^c(s, a) - P(\mathcal{T}^{c_0}(s, a) + \partial_c \mathcal{T}^{c_0}(s, a) \cdot (c - c_0)) \|$$
$$\leq \| \mathcal{T}^c(s, a) - \mathcal{T}^{c_0}(s, a) - \partial_c \mathcal{T}^{c_0}(s, a) \cdot (c - c_0) \|$$
$$+ \| \mathcal{T}^{c_0}(s, a) + \partial_c \mathcal{T}^{c_0}(s, a) \cdot (c - c_0) - P(\mathcal{T}^{c_0}(s, a) + \partial_c \mathcal{T}^{c_0}(s, a) \cdot (c - c_0)) \|$$
$$\leq \| \partial_c^2 \mathcal{T} \|_\infty \| c - c_0 \|^2 + 2 \| \partial_c^2 \mathcal{T} \|_\infty \| c - c_0 \|^2$$
$$\leq 3 \| \partial_c^2 \mathcal{T} \|_\infty \| c - c_0 \|^2.$$

This establishes (16c). $\qquad\square$

Next, we show that the projection map sends nearby measures to nearby probability measures.

**Lemma 14.** *Let $X$ be a metric space, and let $\mu, \nu \in \mathcal{U}(X)$. Then*

$$\| P(\mu) - P(\nu) \| \leq 2 \frac{\| \mu - \nu \|}{\| \mu^+ \|}.$$

*Proof.* By repeatedly applying the triangle inequality, we get

$$\| P(\mu) - P(\nu) \| = \left\| \frac{\mu^+}{\| \mu^+ \|} - \frac{\nu^+}{\| \nu^+ \|} \right\|$$
$$= \left\| \frac{\mu^+ \| \nu^+ \| - \nu^+ \| \mu^+ \|}{\| \mu^+ \| \| \nu^+ \|} \right\|$$
$$\leq \left\| \frac{\mu^+ \| \nu^+ \| - \nu^+ \| \nu^+ \|}{\| \mu^+ \| \| \nu^+ \|} \right\| + \left\| \frac{\nu^+ \| \nu^+ \| - \nu^+ \| \mu^+ \|}{\| \mu^+ \| \| \nu^+ \|} \right\|$$
$$= \frac{\| \mu^+ - \nu^+ \|}{\| \mu^+ \|} + \frac{| \| \nu^+ \| - \| \mu^+ \| |}{\| \mu^+ \|}$$
$$\leq 2 \frac{\| \mu^+ - \nu^+ \|}{\| \mu^+ \|}$$
$$\leq 2 \frac{\| \mu - \nu \|}{\| \mu^+ \|}.$$

$\qquad\square$

The following lemma relates the Wasserstein distance to the total variation distance, and is a special case of a result in the monograph of Villani et al. (2008, Theorem 6.12).

**Lemma 15.** *Let $X$ be a bounded metric space, and let $\mu_1, \mu_2 \in \mathrm{Meas}(X)$. Then*

$$W_1(\mu_1, \mu_2) \leq \mathrm{diam}(X) \| \mu_1 - \mu_2 \|_{\mathrm{Meas}(X)}.$$

*Proof of Theorem 3.* To use Theorem 1, we need to show that the context-enhanced transition and reward functions are good estimates of their true values. By Lemma 13, for all $(s, a) \in \mathcal{S} \times \mathcal{A}$,

$$\|\mathcal{T}^c(s, a) - T^c_{\mathrm{CE}}(s, a)\|_{\mathrm{Meas}(\mathcal{S})} \leq 3\|\partial_c^2\mathcal{T}\|_\infty\|c - c_0\|^2.$$

By Lemma 10, we have

$$W^1(\mathcal{T}^c(s, a), \mathcal{T}^c_{\mathrm{CE}}(s, a)) \leq \mathrm{diam}(\mathcal{S})\|\mathcal{T}^c(s, a) - \mathcal{T}^c_{\mathrm{CE}}(s, a)\|_{\mathrm{Meas}(\mathcal{S})}$$
$$\leq 3\,\mathrm{diam}(\mathcal{S})\|\partial_c^2\mathcal{T}\|_\infty\|c - c_0\|^2.$$

By Taylor's theorem, there exists $c' \in [c, c_0]$ such that

$$|R^c(s, a) - R^{c_0}(s, a) - \partial_c R^{c_0}(s, a) \cdot (c - c_0)| \leq \|\partial_c^2 R^{c'}(s, a)\|\|c - c_0\|^2$$
$$\leq \|\partial_c^2 R\|_\infty\|c - c_0\|^2$$

which we can rewrite as

$$|R^c(s, a) - R^c_{\mathrm{CE}}(s, a)| \leq \|\partial_c^2 R\|_\infty\|c - c_0\|^2$$
$$\|R^c - R^c_{\mathrm{CE}}\|_\infty \leq \|\partial_c^2 R\|_\infty\|c - c_0\|^2.$$

So the reward functions are close.

Next, we will show that the transition and reward functons are Lipschitz. By Lemma 15, for all $(s, a), (s', a') \in \mathcal{S} \times \mathcal{A}$,

$$W^1(\mathcal{T}^c(s, a), \mathcal{T}^c(s', a')) \leq \mathrm{diam}(\mathcal{S})\|\mathcal{T}^c(s, a) - \mathcal{T}^c(s', a')\|_{\mathrm{Meas}(\mathcal{S})}$$
$$\leq \mathrm{diam}(\mathcal{S})L_\mathcal{T} d((s, a), (s', a')),$$

so the Lipschitz constant of $\mathcal{T}^c : \mathcal{S} \times \mathcal{A} \to \mathcal{W}_1(\mathcal{S})$ is at most $\mathrm{diam}(\mathcal{S})L_\mathcal{T}$.

Let us define $\widehat{\mathcal{T}} : \mathcal{S} \times \mathcal{A} \to \mathrm{Meas}(\mathcal{S})$ by

$$\widehat{\mathcal{T}}(s, a) := \mathcal{T}^{c_0}(s, a) + \partial_c\mathcal{T}^{c_0}(s, a) \cdot (c - c_0).$$

Then $\mathcal{T}^c_{\mathrm{CE}} = P \circ \widehat{\mathcal{T}}$. By Lemma 13, for all $(s, a) \in \mathcal{S} \times \mathcal{A}$,

$$\left\|\widehat{\mathcal{T}}(s, a)^+\right\|_{\mathrm{Meas}(\mathcal{S})} \geq 1 - \|\partial_c^2\mathcal{T}\|_\infty\|c - c_0\|^2.$$

For all $(s, a)$ and $(s', a')$ in $\mathcal{S} \times \mathcal{A}$, we have

$$\|\widehat{\mathcal{T}}(s, a) - \widehat{\mathcal{T}}(s', a')\|_{\mathrm{Meas}(\mathcal{S})}$$
$$\leq \|\mathcal{T}^{c_0}(s, a) - \mathcal{T}^{c_0}(s', a')\|_{\mathrm{Meas}(\mathcal{S})} + \|\partial_c\mathcal{T}^{c_0}(s, a) - \partial_c\mathcal{T}^{c_0}(s', a')\|_{\mathrm{Meas}(\mathcal{S})}\|c - c_0\|$$
$$\leq L_\mathcal{T} d((s, a), (s', a')) + L_{\partial_c\mathcal{T}}\|c - c_0\|d((s, a), (s', a')).$$

Then by Lemma 14, for all $(s, a), (s', a') \in \mathcal{S} \times \mathcal{A}$,

$$\left\|P(\widehat{\mathcal{T}}(s, a)) - P(\widehat{\mathcal{T}}(s', a'))\right\|_{\mathrm{Meas}(\mathcal{S})} \leq 2\frac{\|\widehat{\mathcal{T}}(s, a) - \widehat{\mathcal{T}}(s', a')\|_{\mathrm{Meas}(\mathcal{S})}}{\|\widehat{\mathcal{T}}(s, a)^+\|_{\mathrm{Meas}(\mathcal{S})}}$$
$$\leq 2\frac{(L_\mathcal{T} + \|c - c_0\|L_{\partial_c\mathcal{T}})d((s, a), (s', a'))}{1 - \|\partial_c^2\mathcal{T}\|_\infty\|c - c_0\|^2}$$
$$\leq 4(L_\mathcal{T} + \|c - c_0\|L_{\partial_c\mathcal{T}})d((s, a), (s', a')).$$

Again by Lemma 15, we have

$$W^1(\mathcal{T}^c_{\mathrm{CE}}(s, a), \mathcal{T}^c_{\mathrm{CE}}(s', a')) \leq \mathrm{diam}(\mathcal{S})\|\mathcal{T}^c_{\mathrm{CE}}(s, a) - \mathcal{T}^c_{\mathrm{CE}}(s', a')\|_{\mathrm{Meas}(\mathcal{S})}$$
$$\leq 4\,\mathrm{diam}(\mathcal{S})(L_\mathcal{T} + \|c - c_0\|L_{\partial_c\mathcal{T}})d((s, a), (s', a')),$$

so the Lipschitz constant of $\mathcal{T}^c_{\mathrm{CE}} : \mathcal{S} \times \mathcal{A} \to \mathcal{W}_1(\mathcal{S})$ is at most $4\,\mathrm{diam}(\mathcal{S})(L_\mathcal{T} + \|c - c_0\|L_{\partial_c\mathcal{T}})$.

By assumption, $R^c$ is Lipschitz with constant at most $L_R$. The Lipschitz constant of $R^c_{\mathrm{CE}} = R^{c_0} + \partial_c R^{c_0} \cdot (c - c_0)$ is at most $L_R + L_{\partial_c R}\|c - c_0\|$.

Finally, by Theorem 1, we have

$$\|Q^c_{\mathrm{CE}} - Q^c\|_\infty$$
$$\leq \frac{1}{1 - \gamma}\left(\|\partial_c^2 R\|_\infty\|c - c_0\|^2 + \frac{3\gamma\,\mathrm{diam}(\mathcal{S})(1 + L_\pi)\|R\|_{\mathrm{Lip}(\mathcal{S} \times \mathcal{A} \times \mathcal{C})}\|\partial_c^2\mathcal{T}\|_\infty\|c - c_0\|^2}{1 - \gamma\max(1, \mathrm{diam}(\mathcal{S})L_\mathcal{T}(1 + L_\pi))}\right).$$

$\square$

## B.5 CEBE policy is approximately optimal

Here we provide a proof of Theorem 4.

*Proof.* Recall that

$$J(\pi, c) = \mathbb{E}\left[\sum_{t=0}^{\infty} \gamma^t R_t\right] = \mathbb{E}_{s \sim S_0, a \sim \pi(s,c)} Q^c(s, a; \pi),$$

where $R_t$ denotes the reward attained at time $t$ from following the policy $\pi_{\text{CE}}$ in context $c$. Let $c \in \mathcal{C}$. Let $\mu$ denote the initial state distribution. Since $Q_{\text{CE}}$ and $Q_{\text{BE}}$ are close,

$$
\begin{aligned}
J_{\text{CE}}(\pi_{\text{BE}}, c) &= \int_{\mathcal{S}} \int_{\mathcal{A}} Q_{\text{CE}}^c(s, a; \pi_{\text{BE}}) d\pi_{\text{BE}}(a) d\mu(s) \\
&\geq \int_{\mathcal{S}} \int_{\mathcal{A}} Q_{\text{BE}}^c(s, a; \pi_{\text{BE}}) d\pi_{\text{BE}}(a) d\mu(s) \\
&\quad - \int_{\mathcal{S}} \int_{\mathcal{A}} |Q_{\text{CE}}^c(s, a; \pi_{\text{BE}}) - Q_{\text{BE}}^c(s, a; \pi_{\text{BE}})| \, d\pi_{\text{BE}}(a) d\mu(s) \\
&\geq \int_{\mathcal{S}} \int_{\mathcal{A}} Q_{\text{BE}}^c(s, a; \pi_{\text{BE}}) d\pi_{\text{BE}}(a) d\mu(s) - \delta \\
&= J_{\text{BE}}(\pi_{\text{BE}}, c) - \delta.
\end{aligned}
$$

Similarly,

$$
\begin{aligned}
J_{\text{BE}}(\pi_{\text{CE}}, c) &= \int_{\mathcal{S}} \int_{\mathcal{A}} Q_{\text{BE}}^c(s, a; \pi_{\text{CE}}) d\pi_{\text{CE}}(a) d\mu(s) \\
&\geq \int_{\mathcal{S}} \int_{\mathcal{A}} Q_{\text{CE}}^c(s, a; \pi_{\text{CE}}) d\pi_{\text{CE}}(a) d\mu(s) \\
&\quad - \int_{\mathcal{S}} \int_{\mathcal{A}} |Q_{\text{CE}}^c(s, a; \pi_{\text{BE}}) - Q_{\text{BE}}^c(s, a; \pi_{\text{BE}})| \, d\pi_{\text{CE}}(a) d\mu(s) \\
&\geq \int_{\mathcal{S}} \int_{\mathcal{A}} Q_{\text{CE}}^c(s, a; \pi_{\text{CE}}) d\pi_{\text{CE}}(a) d\mu(s) - \delta \\
&= J_{\text{CE}}(\pi_{\text{CE}}, c) - \delta.
\end{aligned}
$$

The above inequalities, combined with the $(\mathcal{C}, \epsilon)$-optimality of $\pi_{\text{BE}}$ and $\pi_{\text{CE}}$ on their respective MDPs, yield

$$
\begin{aligned}
J_{\text{BE}}(\pi_{\text{CE}}, c) &\geq J_{\text{CE}}(\pi_{\text{CE}}, c) - \delta \\
&\geq J_{\text{CE}}(\pi_{\text{BE}}, c) - \epsilon - \delta \\
&\geq J_{\text{BE}}(\pi_{\text{BE}}, c) - \epsilon - 2\delta \\
&\geq \sup_{\rho} J_{\text{BE}}(\rho, c) - 2\delta - 2\epsilon.
\end{aligned}
$$

Hence, $\pi_{\text{CE}}$ is $(\mathcal{C}, 2\delta + 2\epsilon)$-optimal on the Bellman equation. $\qquad\square$

## B.6 Relation of CSE to regularization

In this section we aim to provide intuition about what functions are learned when using CSE by developing a regularization perspective for the CSE loss. For simplicity we consider the supervised setting. In supervised learning, for an input data distribution $\mathcal{D}$ and a true function $F : X \times \mathcal{C} \to Y$ we want to learn, we may define a loss function for a hypothesis $f : X \times \mathcal{C} \to Y$:

$$L^c(f) = \mathbb{E}_{x \sim \mathcal{D}} \|f(x, c) - F(x, c)\|_2^2.$$

The corresponding CSE loss is then obtained by augmenting the data (by adding perturbed inputs and corresponding labels from the linearized true model) and is given by

$$L_{\text{CSE}}^{c_0}(f) = \mathbb{E}_\xi \mathbb{E}_{x \sim \mathcal{D}} \|f(x, c_0 + \xi) - F(x, c_0) - \partial_c F(x, c_0)\xi\|_2^2,$$

where $\xi$ is a random vector of our choice, which we will suppose is standard multivariate Gaussian. We then calculate

$$
\begin{aligned}
&L_{\text{CSE}}^{c_0}(f) \\
&= \mathbb{E}_\xi \mathbb{E}_{x \sim \mathcal{D}} \| f(x, c_0) + \partial_c f(x, c_0)\xi + \xi^T \partial_c^2 f(x, c_0)\xi + O(\|\xi\|^3) - F(x, c_0) - \partial_c F(x, c_0)\xi \|_2^2 \\
&= \mathbb{E}_\xi \mathbb{E}_{x \sim \mathcal{D}} \| f - F \|_2^2 + 2\langle f - F, (\partial_c f - \partial_c F)\xi \rangle \\
&\quad + \| (\partial_c f - \partial_c F)\xi \|_2^2 + \langle \xi^T \partial_c^2 f \xi, f - F \rangle + O(\|\xi\|^3) \\
&= \mathbb{E}_{x \sim \mathcal{D}} \| f - F \|_2^2 + \mathbb{E}_\xi \mathbb{E}_{x \sim \mathcal{D}} \| (\partial_x f - \partial_x F)\xi \|_2^2 + \mathbb{E}_\xi \mathbb{E}_{x \sim \mathcal{D}} \langle \xi^T \partial_c^2 f \xi, f - F \rangle + \mathbb{E}_\xi O(\|\xi\|^3) \\
&\approx \mathbb{E}_{x \sim \mathcal{D}} \| f - F \|_2^2 + \mathbb{E}_\xi \mathbb{E}_{x \sim \mathcal{D}} \| (\partial_x f - \partial_x F)\xi \|_2^2 + \mathbb{E}_\xi \mathbb{E}_{x \sim \mathcal{D}} \langle \xi^T \partial_c^2 f \xi, f - F \rangle.
\end{aligned}
$$

The first term of the above expression encourages $f$ to be close to $F$. The second term is equal to

$$
\begin{aligned}
\mathbb{E}_\xi \mathbb{E}_{x \sim \mathcal{D}} \xi^T (\partial_x f - \partial_x F)^T (\partial_x f - \partial_x F)\xi &= \mathbb{E}_\xi \mathbb{E}_{x \sim \mathcal{D}} \text{Trace}(\xi \xi^T (\partial_x f - \partial_x F)^T (\partial_x f - \partial_x F)) \\
&= \mathbb{E}_{x \sim \mathcal{D}} \text{Trace}(\mathbb{E}[\xi \xi^T](\partial_x f - \partial_x F)^T (\partial_x f - \partial_x F)) \\
&= \sigma^2 \mathbb{E}_{x \sim \mathcal{D}} \text{Trace}((\partial_x f - \partial_x F)^T (\partial_x f - \partial_x F)) \\
&= \sigma^2 \mathbb{E}_{x \sim \mathcal{D}} \| \partial_x f - \partial_x F \|_F^2.
\end{aligned}
$$

and therefore encourages the derivative of $f$ to agree with $F$. Under the assumption that the function $f$ is a close approximation of $F$ in the training context $c_0$, the loss is approximately

$$
L_{\text{CSE}}^{c_0}(f) \approx \sigma^2 \mathbb{E}_{x \sim \mathcal{D}} \| \partial_x f - \partial_x F \|_F^2.
$$

Up to higher-order terms in the perturbation size, data augmentation with CSE corresponds to adding $L^2$ regularization on the difference between the Jacobian of the trained model $f$ and that of the true function $F$. In other words, this encourages $\partial_c f \approx \partial_c F$ at $c_0$.

## C  Additional Experiments

### C.1  PendulumGoal

We provide additional plots for the PendulumGoal experiment. In Figure 5, we show the evaluation performance of the trained policies when sweeping over the context parameters. We additionally plot the mass and length context parameters. All methods perform similarly as we vary the length. When we vary the mass, the methods perform similarly until the mass is greater than about 1.6 at which point the baseline performs best. Note that CSE still outperforms LDR in this case.

### C.2  CartGoal

We present our results on the CartGoal environment in Figure 6. This environment uses discrete action spaces, so we train the $Q$ functions and policies using DQN (Mnih et al., 2013). We can see in the figure that all methods perform similarly with respect to the cart mass and pole mass. CSE has a slight edge on pole mass when the mass is greater than 1.75. Interestingly, baseline performs better than CSE and LDR when changing the gravitational acceleration. We note that this is very out of distribution since the $\Delta c$ used in training was only 0.1. On pole length, the policies trained with LDR performed best and CSE slightly outperforms baseline for most contexts. When varying the goal state, CSE performs best when the goal state is far from the origin.

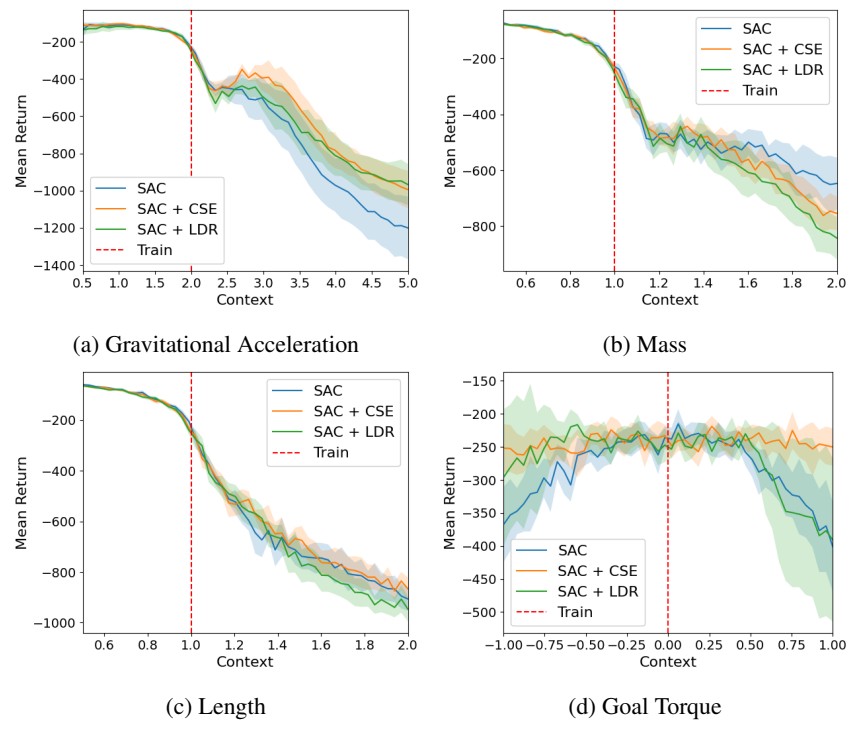

(a) Gravitational Acceleration

(b) Mass

(c) Length

(d) Goal Torque

Figure 5: Comparison of training methods on PendulumGoal as we vary the context parameters.

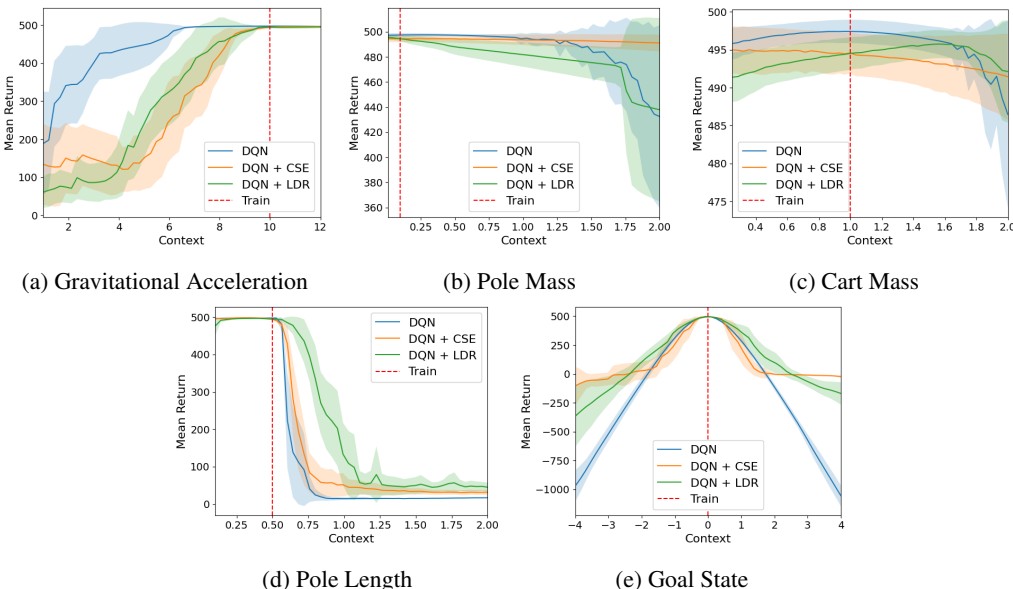

(a) Gravitational Acceleration

(b) Pole Mass

(c) Cart Mass

(d) Pole Length

(e) Goal State

Figure 6: Comparison of training methods on CartGoal as we vary the context parameters.

## C.3 AntGoal

We present the results of the AntGoal environment in Figure 7. We can see that CSE performs at least as well as LDR across all contexts. Both CSE and LDR beat baseline performance across all contexts.

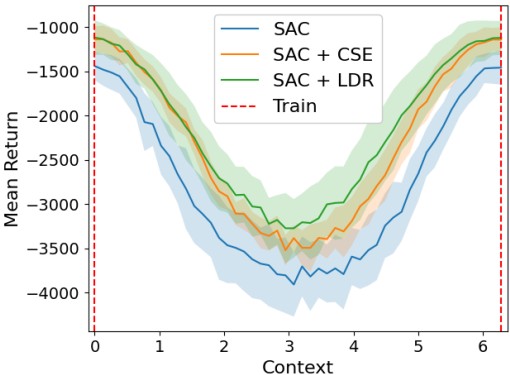

Figure 7: Comparison of training methods as we vary angle of the goal state in the AntGoal environment and keep a fixed distance of 3. The training context is $(3.0, 0.0)$.

### C.4 PendulumGoal with Automatic Differentiation

In the other experiments, we use analytical gradients, but this may be limiting in some settings. In this section, we perform an experiment on the PendulumGoal environment and use automatic differentiation to obtain gradients. Figure 8 shows the evaluation performance of the trained policies when sweeping over context parameters in the PenGoal-AD environment. We can see that CSE performs similarly to LDR. In some cases, CSE outperforms LDR (e.g., when gravitational acceleration is at least 3).

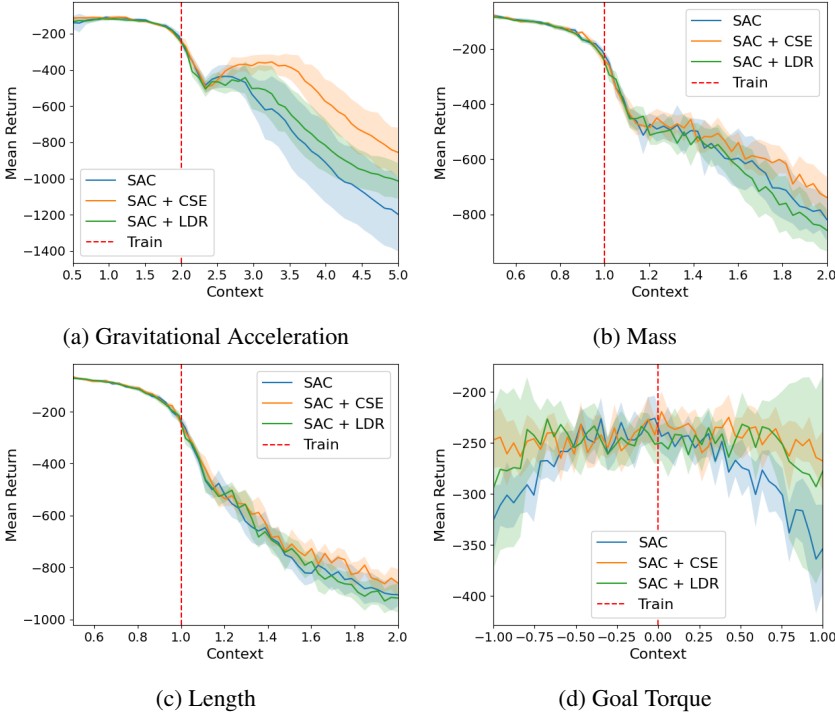

(a) Gravitational Acceleration

(b) Mass

(c) Length

(d) Goal Torque

Figure 8: Comparison of training methods on PendulumGoal-AD as we vary the context parameters.

### C.5 Stochastic Transitions

We present the results for the SimpleDirection-Stochastic environment in Figure 9. This environment has the same transitions and rewards as in SimpleDirection, but with additive Gaussian noise $\epsilon \sim$

$\mathcal{N}(0, 0.1)$ applied to both the transitions and rewards at each step. In this environment, the policy trained with CSE has similar performance as with LDR and outperforms the baseline. While this provides empirical evidence that CSE is robust to stochasticity in transitions and rewards, further work should establish theoretical guarantees for this scenario.

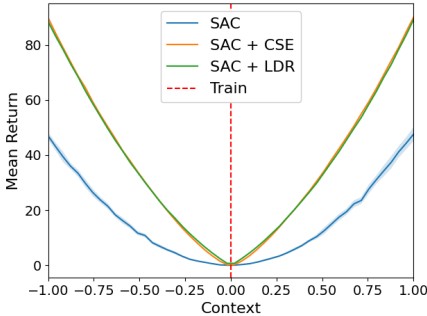

Figure 9: Comparison of training methods as we vary the first context parameter in SimpleDirection-Stochastic.

## C.6  Aggregate Statistics and Context Dimension

In this section, we present a table of average performance for each environment. We compute the mean performance from the previous context sweeps and show the results in Table 1. The last column shows the normalized score

$$\frac{\text{CSE} - \text{Baseline}}{\text{LDR} - \text{Baseline}}$$

for each environment. Note that CSE is better than the baseline in all cases. CSE performs better than LDR in three of the six environments tested (CheetahVelocity, SimpleDirection, and PendulumGoal). The data do not point to a relationship between the context dimension and the normalized score. This is expected since the transitions and rewards vary significantly between environments. We expect that the context dimension can have an impact on the performance of LDR and CSE and further work should consider testing this when scaling to high dimensional contexts.

| Environment | Dimension($\mathcal{C}$) | Baseline | LDR | CSE | Normalized Score |
|---|---|---|---|---|---|
| CheetahVelocity | 1 | $-681.87$ | $-588.22$ | $\mathbf{-575.94}$ | 1.13 |
| SimpleDirection | 2 | 16.03 | 37.60 | $\mathbf{38.05}$ | 1.02 |
| AntDirection | 2 | 507.11 | $\mathbf{560.23}$ | 515.98 | 0.17 |
| AntGoal | 2 | $-2790.55$ | $\mathbf{-2157.52}$ | $-2307.24$ | 0.76 |
| PendulumGoal | 4 | $-425.08$ | $-421.58$ | $\mathbf{-394.19}$ | 8.83 |
| CartGoal | 5 | 284.65 | $\mathbf{325.49}$ | 315.74 | 0.76 |

Table 1: Mean Returns averaged across all context sweeps for each environment using analytic gradients. The last column shows the normalized score for CSE compared to LDR. The highest mean returns for each environment are shown in bold.

## D  Experiment Details

In this section, we outline the details of our experimental setup. We include an overview of the SAC hyperparameters in Table 2 and replay buffer hyperparameters in Table 3. For the SAC experiments, all neural networks use three fully connected layers of width 256 with ReLU activations. For the CartGoal experiment, we use DQN and the neural networks have a single fully connected layer of width 256 with Tanh activations. The neural network head is a two layer fully connected neural network with width 256 and ReLU activations. We use the default RLLib implementations of SAC and default hyperparameters for each environment are based on the tuned examples from RLLib

| Environment | Epochs | Actor LR | Critic LR | Entropy LR | Initial Entropy |
|---|---|---|---|---|---|
| SimpleDirection | 500 | $1 \cdot 10^{-3}$ | $2 \cdot 10^{-3}$ | $4 \cdot 10^{-4}$ | 1.0 |
| PendulumGoal | 1000 | $2 \cdot 10^{-4}$ | $8 \cdot 10^{-4}$ | $9 \cdot 10^{-4}$ | 1.001 |
| CheetahVelocity | 10000 | $2 \cdot 10^{-4}$ | $8 \cdot 10^{-4}$ | $9 \cdot 10^{-4}$ | 1.001 |
| AntDirection | 5000 | $3 \cdot 10^{-5}$ | $3 \cdot 10^{-4}$ | $1 \cdot 10^{-4}$ | 1.001 |
| AntGoal | 10000 | $3 \cdot 10^{-5}$ | $3 \cdot 10^{-4}$ | $1 \cdot 10^{-4}$ | 0.01 |

Table 2: SAC Hyperparameters

| Environment | Capacity | $\alpha$ | $\beta$ |
|---|---|---|---|
| SimpleDirection | $1e6$ | 0.6 | 0.4 |
| PendulumGoal | $1e5$ | 1.0 | 0.0 |
| CartGoal | $5e4$ | 0.6 | 0.4 |
| CheetahVelocity | $1e5$ | 0.6 | 0.4 |
| AntDirection | $1e6$ | 0.6 | 0.4 |
| AntGoal | $1e6$ | 0.6 | 0.4 |

Table 3: Prioritized Episode Replay Buffer Hyperparameters

(Liang et al., 2018, 2021). Note that Lee and Chung (2021) use a smaller initial entropy penalty for AntGoal, so we use their choice of $\alpha_0 = 0.02$ as well. We increase the number of epochs in each environment to account for the increased complexity of training with contexts. When performing CSE and LDR, we generate perturbations $\Delta c$ uniformly from the sphere of radius 0.1. To improve the sampling performance, we vectorize all environments to sample from 8 versions of the environment.

When training with SAC, we have the following additional hyperparameters which we keep constant across experiments. The target entropy is automatically tuned by RLLib. The policy polyak averaging coefficient is $5 \cdot 10^{-3}$. We use a training batch size of 256 and 4000 environment steps per training epoch. These are the default choices in RLLib. For CartGoal, we also have the following hyperparameters with DQN. The learning rate is $5 \cdot 10^{-4}$ and the train batch size is 32. We additionally enable double DQN and dueling within DQN to improve $Q$ value estimation, implemented within Liang et al. (2018, 2021). We also use random exploration for $10^4$ timesteps, followed by using $\epsilon$-greedy exploration with $\epsilon = 0.02$. These are hyperparameters were chosen based on the default RLLib implementations DQN for the tuned CartPole example from RLLib (Liang et al., 2018, 2021). Note that in both SAC and DQN, we use 1-step returns during training. While the experiments in this paper use SAC and DQN, we note that CSE is derived from CEBE which should allow one to apply the method to any algorithm which leverages bootstrapping. Issues may arise when using $n$-step returns for estimating returns as repeated linear approximations will accumulate error for large $n$.

**Hardware** Experiments were run on a system with Intel(R) Xeon(R) Gold 6152 CPUs @ 2.10GHz and NVIDIA GeForce RTX 2080 Ti GPUs.

**Python libraries** Torch (Paszke et al., 2019); Ray (Moritz et al., 2018); Ray Tune (Liaw et al., 2018); Ray RLlib (Liang et al., 2018, 2021); Seaborn (Waskom, 2021); Matplotlib (Hunter, 2007); Pandas (pandas development team, 2020); Numpy (Harris et al., 2020); Scipy (Gommers et al., 2024); Mujoco (Todorov et al., 2012); Gymnasium (Towers et al., 2024).

# E   Environments

We provide additional details about the environments in this section. A summary of the environment parameters is included in Table 4.

| Environment | Train Context | $\gamma$ | Horizon |
|---|---|---|---|
| SimpleDirection | $(0.0, 0.0)$ | 0.9 | 10 |
| PendulumGoal | $(2.0, 1.0, 1.0, 0.0)$ | 0.99 | 200 |
| CartGoal | $(10.0, 0.1, 1.0, 0.5, 0.0)$ | 0.99 | 500 |
| CheetahVelocity | $(2.0)$ | 0.99 | 1000 |
| AntDirection | $(1.0, 0.0)$ | 0.99 | 1000 |
| AntGoal | $(1.0, 0.0)$ | 0.99 | 1000 |

Table 4: Environment Parameters

### E.1  Cliffwalking

We begin our experiments by first solving the problem in a tabular setting. This allows us to exactly solve the Bellman equation and the CEBE to avoid noise due to sampling and training in deep RL. This allows us to compute the approximation error exactly. We consider the tabular Cliffwalking environment shown in Figure 10. In this environment, the agent must navigate to a goal state without first falling off a cliff (depicted as black cells in the figure). If the agent walks near the cliff's edge it can complete the task faster, but risks falling off the cliff. In this environment, the context is $c \in [0, 1]$, which denotes the probability that the agent slips. When the agent slips, the next state is picked uniformly at random among the adjacent states. If the slipping probability is high, then the agent is incentivized to first navigate away from the edge of the cliff before navigating to the goal state.

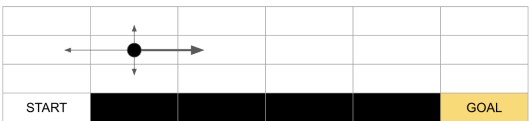

Figure 10: Visualization of the tabular Cliffwalker environment. The agent, represented by ●, is taking the `right` action. The agent is most likely to move right, but there is a probability $c$ of the agent slipping. If the agent slips to moves to an adjacent cell with equal probability.

### E.2  SimpleDirection

Recall that the equations for the SimpleDirection environment are given by

$$\mathcal{T}^c(s, a) = s + a + c, \quad R^c(s, a, s') = s' \cdot c.$$

We pick this environment because the transition is linear in each variable and rewards are linear in $c$. The spaces are

$$\mathcal{S} = \mathbb{R}^2, \quad \mathcal{A} = [-1, 1]^2, \quad \mathcal{C} = [-1, 1]^2,$$

and the initial state is picked uniformly at random $s_0 \sim \text{Uniform}\left([-1, 1]^2\right)$. This gives the resulting gradients

$$\frac{\partial \mathcal{T}^c}{\partial c} = I, \quad \frac{\partial R^c}{\partial c} = s', \quad \frac{\partial R^c}{\partial s'} = c.$$

### E.3  PendulumGoal

The transition function is the same as in the pendulum environment from Towers et al. (2024) and are determined by the solution to the ODE

$$\ddot{\theta} = \frac{3g}{2l} \sin \theta + \frac{3}{ml^2} u$$

over a time interval $dt = 0.02$. We modify the reward so that the desired position of the pendulum is not vertical. In particular, we specify $\tau$ to be the desired torque applied at the goal state. We choose to specify a goal torque instead of a goal state since the action space is bounded and an arbitrary goal state may not be stable for any policy under different contexts (e.g., when $\max |u| < \frac{mgl}{2} \sin(\theta_{\text{goal}})$). Solving for the goal state at the desired torque $\tau$, we get

$$\theta_{\text{goal}} = \arcsin\left(-\frac{2\tau}{mgl}\right).$$

The reward function is then

$$R^c = \pi^2 \sin\left(\frac{\theta_{\text{goal}} - \theta}{2}\right)^2 + 0.1\,\dot{\theta}^2 + 0.001\,u^2,$$

where we have also used $\sin$ instead of the absolute value function for the state error due to periodicity in the problem.

### E.4 CartGoal

The transition function is the same as in the cartpole environment from Towers et al. (2024) and are determined by the solution to the system of ODEs

$$\ddot{\theta} = \frac{g\sin\theta + \cos\theta\left(\frac{-f - m_p l\dot{\theta}^2 \sin\theta}{m_p + m_c}\right)}{l\left(\frac{4}{3} - \frac{m_p \cos^2\theta}{m_p + m_c}\right)}$$

$$\ddot{x} = \frac{f + m_p l\left(\dot{\theta}^2 \sin\theta - \ddot{\theta}\cos\theta\right)}{m_p + m_c}$$

over a time interval $dt = 0.02$. The reward function is modified to include a goal state for the cart position

$$R^c = 2 - \sqrt{1 + (x - x_{\text{goal}})^2}.$$

Note that the environment uses the same termination conditions as CartPole and can terminate early if $x \notin [-2.4, 2.4]$ or $\theta \notin [-0.21, 0.21]$.

### E.5 ODE Environment Simulation

The continuous control environments are specified by differential equations and reward functions. Derivatives of $\mathcal{T}^c$ and $R^c$ are computed symbolically. The differential equations are then converted into first-order systems and compiled into Python functions before solving numerically with Euler's method.

**Example with a simple equation** Consider the transition function specified by the differential equation

$$\dot{x} = ax + bF$$

and reward function

$$R^{(a,b,c)} = -(c - x(t + \Delta t))^2,$$

where $a, b, c \in \mathbb{R}$ are context variables and $F$ is the control. Then the derivatives of the transition function with respect to the context variables are solutions of the following differential equations:

$$\frac{d\dot{x}}{da} = a\frac{dx}{da} + x, \quad \frac{d\dot{x}}{db} = a\frac{dx}{db} + F, \quad \frac{d\dot{x}}{dc} = a\frac{dx}{dc} = 0.$$

Similarly, the reward function has derivatives

$$\frac{dR^{(a,b,c)}}{da} = 0, \quad \frac{dR^{(a,b,c)}}{db} = 0, \quad \frac{dR^{(a,b,c)}}{dc} = -2(c - x), \quad \frac{dR^{(a,b,c)}}{dx} = 2(c - x).$$

When solving for the gradients, we use the initial conditions

$$\frac{dx}{da}(0) = 0, \quad \frac{dx}{db}(0) = 0, \quad \frac{dx}{dc}(0) = 0.$$

This is because we care about how the next state would change as we change the context. So, the change in the next state due to the change in the context would be zero if we did not move forward in time at all. We include a full derivation for the PendulumGoal system in Appendix F.2.

### E.6 Goal-based Mujoco Environments

Lee and Chung (2021) introduced goal-based variants of the HalfCheetah and Ant from Mujoco (Todorov et al., 2012). We denote these environment by CheetahVelocity, AntDirection, and AntGoal. They each follow the same dynamics equations specified by Todorov et al. (2012). However, their rewards are modified to introduce context dependent tasks.

The original CheetahVelocity implementation modifies the forward reward to a penalty on the velocity $-|v - v_{\text{goal}}|$. We use a smoothed version of this function,

$$R^c_{\text{Velocity}} = 1 - \sqrt{1 + (v - v_{\text{goal}})^2}.$$

We follow the original reward function for the AntDir environment. The velocity component of the Ant reward function is modified to

$$R^c_{\text{Direction}} = c \cdot v.$$

The original AntGoal implementation modifies the forward reward to a goal based penalty $-|(x, y) - (x_{\text{goal}}, y_{\text{goal}})|$. We use a smoothed version of this function,

$$R^c_{\text{Goal}} = 1 - \sqrt{1 + ((x, y) - (x_{\text{goal}}, y_{\text{goal}}))^2}.$$

## F Worked Examples

In this section, we provide fully worked out calculations for some of the environments in the paper with the aim of providing more clarity.

### F.1 SimpleDirection expected optimal returns

The $k$th state is given by

$$s_k = s_0 + \sum_{i=0}^{k-1} a_i + kc.$$

Using this, we get the rewards are

$$r_k = c \cdot s_{k+1} = c \cdot s_0 + c \cdot \sum_{i=0}^{k} a_i + (k+1)\|c\|_2^2.$$

By symmetry on $s_0$ initial distribution, we have

$$\mathbb{E}(r_k) = c \cdot \sum_{i=0}^{k} a_i + (k+1)\|c\|_2^2.$$

Due to the constraints on $a_i$, the maximum is obtained when $a_i = \text{sign}(c)$ (computed element-wise). So, we get

$$\mathbb{E}(r_k) = (k+1)\left(\|c\|_1 + \|c\|_2^2\right).$$

The expected non-discounted returns are thus

$$\sum_{k=0}^{H-1} r_k = \sum_{k=0}^{H-1} (k+1)\left(\|c\|_1 + \|c\|_2^2\right) = \binom{H+1}{2}\left(\|c\|_1 + \|c\|_2^2\right),$$

where $H$ is the horizon.

### F.2 PendulumGoal Equations

In this section, we present the full set of equations for the PendulumGoal Environment. This includes the transition and reward gradients. Recall that the context parameters are $(g, m, l, \tau)$, where $g$ is the

gravitational constant, $m$ is mass, $l$ is the pendulum length, and $\tau$ is the goal torque. The dynamics are governed by the equation

$$\ddot{\theta} = \frac{3g}{2l} \sin(\theta) + \frac{3}{ml^2} u,$$

where $\theta$ is the pendulum angle and $u$ is the control torque. The reward function is

$$R^c = \pi^2 \sin\left(\frac{\theta_{\text{goal}} - \theta}{2}\right)^2 + 0.1\,\dot{\theta}^2 + 0.001\,u^2,$$

where $\theta_{\text{goal}} = \sin^{-1}\left(\frac{-2\tau}{mgl}\right)$. Taking gradients of the transition dynamics in each context parameter, we get

$$\ddot{\theta}_g = \frac{3}{2l}\left(\sin(\theta) + g\cos(\theta)\theta_g\right)$$

$$\ddot{\theta}_m = \frac{3g}{2l}\cos(\theta)\theta_m - \frac{3}{m^2 l^2}u$$

$$\ddot{\theta}_l = -\frac{3g}{2l^2}\sin(\theta) + \frac{3g}{2l}\cos(\theta)\theta_l - \frac{6}{ml^3}u$$

$$\ddot{\theta}_\tau = \frac{3g}{2l}\cos(\theta)\theta_\tau.$$

The full transition dynamics with the gradient equations are then converted into the following first order system of differential equations.

$$\dot{\phi} = \frac{3g}{2l}\sin(\theta) + \frac{3}{ml^2}u$$

$$\dot{\theta} = \phi$$

$$\dot{\phi}_g = \frac{3}{2l}\left(\sin(\theta) + g\cos(\theta)\theta_g\right)$$

$$\dot{\theta}_g = \phi_g$$

$$\dot{\phi}_m = \frac{3g}{2l}\cos(\theta)\theta_m - \frac{3}{m^2 l^2}u$$

$$\dot{\theta}_m = \phi_m$$

$$\dot{\phi}_l = -\frac{3g}{2l^2}\sin(\theta) + \frac{3g}{2l}\cos(\theta)\theta_l - \frac{6}{ml^3}u$$

$$\dot{\theta}_l = \phi_l$$

$$\dot{\phi}_\tau = \frac{3g}{2l}\cos(\theta)\theta_\tau$$

$$\dot{\theta}_\tau = \phi_\tau,$$

where $\phi = \frac{d\theta}{dt}$. The reward function and its gradients are

$$R^c = \pi^2 \sin\left(\frac{\theta_{\text{goal}} - \theta}{2}\right)^2 + 0.1\,\phi^2 + 0.001\,u^2$$

$$R^c_g = \pi^2 \cos\left(\frac{\theta_{\text{goal}} - \theta}{2}\right) \sin\left(\frac{\theta_{\text{goal}} - \theta}{2}\right)(\partial_g\theta_{\text{goal}} - \theta_g) + 0.2\,\phi_g\phi$$

$$R^c_m = \pi^2 \cos\left(\frac{\theta_{\text{goal}} - \theta}{2}\right) \sin\left(\frac{\theta_{\text{goal}} - \theta}{2}\right)(\partial_m\theta_{\text{goal}} - \theta_g) + 0.2\,\phi_m\phi$$

$$R^c_l = \pi^2 \cos\left(\frac{\theta_{\text{goal}} - \theta}{2}\right) \sin\left(\frac{\theta_{\text{goal}} - \theta}{2}\right)(\partial_l\theta_{\text{goal}} - \theta_l) + 0.2\,\phi_l\phi$$

$$R^c_\tau = \pi^2 \cos\left(\frac{\theta_{\text{goal}} - \theta}{2}\right) \sin\left(\frac{\theta_{\text{goal}} - \theta}{2}\right)(\partial_\tau\theta_{\text{goal}} - \theta_l) + 0.2\,\phi_\tau\phi,$$

where

$$\partial_g \theta_{\text{goal}} = \frac{2\tau}{mg^2 l \sqrt{1 - \frac{4\tau^2}{m^2 g^2 l^2}}}$$

$$\partial_m \theta_{\text{goal}} = \frac{2\tau}{m^2 gl \sqrt{1 - \frac{4\tau^2}{m^2 g^2 l^2}}}$$

$$\partial_l \theta_{\text{goal}} = \frac{2\tau}{mgl^2 \sqrt{1 - \frac{4\tau^2}{m^2 g^2 l^2}}}$$

$$\partial_\tau \theta_{\text{goal}} = \frac{-2}{mgl \sqrt{1 - \frac{4\tau^2}{m^2 g^2 l^2}}}.$$

## G   Broader Impacts

This paper develops foundations for generalization in RL with sufficiently regular CMDPs. This may be applied to improve performance in out-of-distribution contexts. Improvements in this direction aid in increasing our understanding of CMDPs and can lead to the design of policies which are more robust to perturbations of the context. This work focuses on sufficiently smooth CMDPs, which appear, for instance, in physics problems. The authors do not foresee any direct negative societal impacts of this work, but do not have control over how these methods might be applied in practice.

