# OpenReview forum: "Zero-Shot Context Generalization in Reinforcement Learning from Few Training Contexts"
_NeurIPS.cc/2025/Conference — NeurIPS 2025 poster_

### Official Review · Reviewer_W69A · 2025-06-14

**Clarity:** 3
**Significance:** 2
**Originality:** 2
**Rating:** 5
**Confidence:** 2

**Summary:**

CLAIMS:

Proposed CEBE based on the perturbation over CMDPs: The analyses surrounding how CEBE helps OOD generalization are technically sound, despite quite strong structural assumptions;

Derived CSE as a data augmentation approach to approximate CEBE (for deterministic control): CSE is simple and straightforward, as presented in Sec. 4;

The authors claim that CEBE can help agents' generalization by training even on a single task. CSE is validated in experiments with 1-D contexts: 1-D contexts missed the opportunity to better convince the readers about the scalability and the potential of the proposed method, as discussed below;

NOVELTY:

the CSE method is simple and straightforward, and possibly can be combined with other techniques to yield better generalization

CSE seems to work well with even stochastic rewards, despite not mathematically proven

**Questions:**

Zhao et al. 2020 was inappropriately cited in Sec. 5 line 223. Domain randomization was indeed mentioned in the survey, but authors should instead properly acknowledge the origin of the method, instead of citing a survey that mentioned the method, let alone the fact the survey was already cited several times before this.

**Ethical Concerns:**

["NO or VERY MINOR ethics concerns only"]

**Final Justification:**

The authors corrected my misunderstanding about the dimensionality of the context variables, which addressed my concern regarding the work's scalability measurably.

**Limitations:**

The experimental contexts seem to be all 1D. Any additional experiment that goes beyond scalar context values could be a powerful argument for the scalability of the method.

I think figure 1 can be better designed / scaled. Currently, it takes too much space in the main manuscript and it shows quite little.

The work also would benefit from a comparison with "Consciousness-Inspired Spatio-Temporal Abstractions for Better Generalization in Reinforcement Learning" (by Bengio's group), which heavily used discrete combinatorial contexts to demonstrate the generalization abilities of agents and the scalability of the methods.

**Quality:**

3

**Strengths And Weaknesses:**

The assumptions of explicit and differentiable context limit the applicability of the method.

---

> ### Author Rebuttal · Authors · 2025-07-31
>
> We thank the reviewer for their valuable comments, questions and suggestions that will improve the final manuscript.
>
> > The experimental contexts seem to be all 1D. Any additional experiment that goes beyond scalar context values could be a powerful argument for the scalability of the method.
>
> The experiments use higher-dimensional context spaces during training. However, plotting the performance of this against all the dimensions is not possible for a paper. Training is performed with the full dimensionality of the context space, but the plots are generated by performing a sweep over a single dimension. This allows us to view slices of the evaluation performance as we vary the context. In Table 3 from Appendix E, we list details about the training context. We have included the context space dimensions for each environment here for convenience.
>
> | Environment     | Context Space Dimension |
> |-----------------|------------------------:|
> | Cliffwalker     |                       1 |
> | SimpleDirection |                       2 |
> | PendulumGoal    |                       4 |
> | CheetahVelocity |                       1 |
> | AntDirection    |                       2 |
> | CartGoal        |                       5 |
> | AntGoal         |                       2 |
>
> > CSE seems to work well with even stochastic rewards, despite not mathematically proven
>
> Our work focuses primarily on deterministic rewards, but the stochastic reward setting is an important case to consider for future works. We note that the methods used in this paper to address stochastic transitions should naturally extend to stochastic rewards. We ran an additional experiment for SimpleDirection with additive Gaussian noise in the transitions and rewards. The results are similar to the SimpleDirection results with CSE performance matching LDR, both of which outperform Baseline. We will include these results in the final manuscript.
>
> Similar to the transition function, the structure of the reward function and choice of approximation can influence the overall results. Future works should leverage the structure of the underlying CMDP, domain constraints, and available reward gradient information to motivate these decisions. For example, in simulated environments, one can use the reparameterization trick in combination with automatic differentiation so that the simulator can provide reward gradients. We agree that extending the theory in this direction is a valuable avenue for future work. While we expect the generalization to carry over naturally, we focused on the deterministic case to maintain a clear presentation of the core ideas.
>
> > Zhao et al. 2020 was inappropriately cited in Sec. 5 line 223. Domain randomization was indeed mentioned in the survey, but authors should instead properly acknowledge the origin of the method, instead of citing a survey that mentioned the method, let alone the fact the survey was already cited several times before this.
>
> Thank you for pointing this out. The concept of domain randomization has existed since the 1990s and can be attributed to multiple sources. We will add the following sources to the final manuscript:
>
> - Nick Jakobi, Phil Husbands, and Inman Harvey. “Noise and the reality gap: The use of simulation in evolutionary robotics”. In: European Conference on Artificial Life. Springer. 1995, pp. 704–720.
> - Ofir Levy and Lior Wolf. “Live repetition counting”. In: Proceedings of the IEEE International Conference on Computer Vision. 2015, pp. 3020–3028.
> - Fereshteh Sadeghi and Sergey Levine. “(CAD)ˆ2 RL: Real Single-Image Flight without a Single Real Image”. In: arXiv preprint arXiv:1611.04201 (2016).
> - Josh Tobin, Rachel Fong, Alex Ray, Jonas Schneider, Wojciech Zaremba, and Pieter Abbeel. Domain randomization for transferring deep neural networks from simulation to the real world. In 2017 IEEE/RSJ international conference on intelligent robots and systems (IROS), pp. 23–30. IEEE, 2017.
>
> If there are any other sources that you would recommend, please let us know.
>
> > I think figure 1 can be better designed / scaled. Currently, it takes too much space in the main manuscript and it shows quite little.
>
> Thank you for this comment. We added this figure to establish a direct connection between the theory and experiments. We will make this figure smaller in the final manuscript.
>
> > The work also would benefit from a comparison with "Consciousness-Inspired Spatio-Temporal Abstractions for Better Generalization in Reinforcement Learning" (by Bengio's group), which heavily used discrete combinatorial contexts to demonstrate the generalization abilities of agents and the scalability of the methods.
>
> Thank you for suggesting this source. The work on options and hierarchical planning is very interesting and important, especially in the context of generalization over causal reasoning tasks. This work uses hierarchical planning to generalize task information across variations in gridworld environments. For example, an agent may need to first find a key to unlock a door before navigating to a goal state. The agent must learn spatial abstraction to handle changing key locations, as well as temporal abstraction to learn the causal relationship between the subtasks in the problem. The authors design a hierarchical policy which achieves better generalization on out-of-distribution tasks. It is important to note that this depends on the number of contexts (seeds) used during training. In figure 4 of their paper, we see that out-of-distribution generalization increases significantly when at least 25 training contexts are used.
>
> Our work leverages approximations of the CMDP to improve zero-shot, out-of-distribution generalization. By using gradient information, we can improve generalization with only one training context using data augmentation. The focus of their work is in designing a hierarchical policy while our work is independent of the choice of architecture. Importantly, the flexibility of our method allows one to incorporate CEBE and/or CSE into other frameworks to make improvements. The authors of this work wrote their code only for gridworld environments (discrete state and action spaces) and noted challenges of extending this to continuous problems (e.g., goal identification). As you noted, the context spaces they use are discrete whereas ours are continuous. These factors make it challenging to experimentally compare with our work. An interesting future direction may combine our work with theirs to account for combinations of discrete and continuous context variables.
>
> We will discuss this in the final version of the manuscript.

---

> ### Comment · Reviewer_W69A · 2025-08-03
>
> Sorry for my previous misunderstanding of the dimensionality of the context variables!
>
> If possible, can you add some more explicit explanation to, for example, the captions?
>
> Maybe, design some figures that can show the dimensionality of context variables on the x-axes and the performance on the y-axes, to demonstrate the scalability of your approach, similar to what was done in the Bengio's group's work?
>
> Thanks!

---

> > ### Author Response · Authors · 2025-08-06
> >
> > Thank you again for your feedback and suggestions. Yes, we will make this more explicit in the paper by adding a note to the captions of the figures. We will also add a figure which compares the dimensionality of the context space with the generalization performance of the methods.

---

> > > ### Comment · Reviewer_W69A · 2025-08-06
> > >
> > > Thank you!!

---

### Official Review · Reviewer_dZY9 · 2025-07-02

**Clarity:** 3
**Significance:** 3
**Originality:** 3
**Rating:** 4
**Confidence:** 3

**Summary:**

The paper proposes a method to enhance zero-shot generalisation of RL methods by exploiting regularity in the set of environments the policies are trained in. The regularity is modelled through contextual markov decision process (CMDPs), and specifically regularity in the reward and transition functions wrt. the context parameters. The work introduces a contex-enhanced bellman equation (CEBE) which improves generalisation from training in a single context by updating the model on similar contexts through exploiting the regularity. The effectiveness of CEBE is demonstrated theoretically and empirically, and a data augmentation method (CSE) is derived from CEBE that approximates CEBE in control environments. CSE is compared empirically to baselines in these environments and performs well.

**Questions:**

covered above.

**Ethical Concerns:**

["NO or VERY MINOR ethics concerns only"]

**Final Justification:**

My original review serves as the justification for my score. The authors provided several more results, but I do not see them as substantial enough to raise my score from 4 to 5. I'm still happy for the paper to be accepted.

**Limitations:**

yes

**Paper Formatting Concerns:**

no concerns

**Quality:**

2

**Strengths And Weaknesses:**

# Strengths

* The paper is well-written, clear and easy to understand
* The theoretical work is clear and an interesting contribution
* the topic of the work is important and significant, the contribution novel, and the methodology for improving generalisation interesting and promising.
* The empirical validation of the theory in figure 1 is beneficial, and the method being relatively close to the theoretical one is also beneficial in aiding understanding of the method's success.

# Weaknesses

The first weakness of the contribution is in the somewhat minimal empirical justification of the method's benefits. Relatively few environments are experimented on, and only 1 meaningful baseline is compared too, which makes it difficult to assess whether the proposed method is an improvement over the current state-of-the-art in this space. A more thorough comparison against baselines in a wider set of environments would be beneficial.

A second weakness is that the method relies on having direct access to the context and to the derivates of the reward and transition function wrt. the context. These assumptions are quite limiting as to the applicability of the method, and hence it would be beneficial to discuss whether they can be mitigated through further approximations, or whether they are fundamental to the method.

# Summary

Overall, I think the paper is just worthy of acceptance currently. Addressing the two weaknesses above would help me raise my score.

---

> ### Author Rebuttal · Authors · 2025-07-31
>
> We thank the reviewer for their valuable comments and suggestions that will improve the final manuscript.
>
> > The first weakness of the contribution is in the somewhat minimal empirical justification of the method's benefits. Relatively few environments are experimented on, and only 1 meaningful baseline is compared too, which makes it difficult to assess whether the proposed method is an improvement over the current state-of-the-art in this space. A more thorough comparison against baselines in a wider set of environments would be beneficial.
>
> In this work, we conduct experiments across several regimes to demonstrate the benefits and generality of our method. Specifically, we validate Theorem 3 using the Cliffwalker environment, providing a clear link between theory and practice. To assess CSE in discrete control settings, we evaluate it on the CartGoal environment using the DQN algorithm. For continuous control, we test CSE on a diverse set of benchmark environments, including SimpleDirection, PendulumGoal, CheetahVelocity, AntVelocity, and AntDirection. These environments were chosen to cover a range of dynamics, context structures, and action spaces. We ran additional experiments using automatic differentiation and stochastic transitions and rewards, which we will add to the final manuscript.
>
> We compare against domain randomization, as it is a widely adopted and well-established baseline for out-of-distribution generalization in RL. Notably, domain randomization improves generalization by leveraging exact data from nearby contexts, whereas CSE approximates transitions and rewards in these regions. In this sense, domain randomization can be viewed as an exact counterpart to CSE. Both approaches emphasize the influence of data, rather than other factors such as model architecture. This enables a fair comparison across a diverse set of environments while keeping implementation details consistent between methods.
>
> > A second weakness is that the method relies on having direct access to the context and to the derivates of the reward and transition function wrt. the context. These assumptions are quite limiting as to the applicability of the method, and hence it would be beneficial to discuss whether they can be mitigated through further approximations, or whether they are fundamental to the method.
>
> The theory developed for CSE assumes exact gradients to simplify the presentation. However, sufficiently accurate estimates for the gradients should suffice for CSE. One may use differentiable simulators like Brax to take gradients through the simulation using automatic differentiation. One can also use finite difference approximations, but this requires that one simulates $|\mathcal{C}| + 1$ contexts. We ran an additional experiment using automatic differentiation with the PendulumGoal environment and present the mean performance across each context sweep here:
>
> |          | Gravity | Mass    | Length  | Goal Torque |
> |----------|---------|---------|---------|-------------|
> | Baseline | -541.48 | -423.61 | -502.22 | -273.15     |
> | LDR      | -496.67 | -440.14 | -503.97 | -252.42     |
> | CSE      | -388.47 | -398.71 | -477.59 | -245.40     |
>
> In each sweep across a context variable, CSE performs better than the baseline and LDR. We will add a plot corresponding to this experiment accompanied by the above discussion in the final manuscript.

---

> > ### Comment · Reviewer_dZY9 · 2025-08-04
> >
> > I thank the authors for their response.
> >
> > On this point
> > > We ran additional experiments using automatic differentiation and stochastic transitions and rewards, which we will add to the final manuscript.
> >
> > You have shared the automatic differentiation results, but can you also share the stochastic transition/reward results here as well? Those results would be beneficial to understanding the proposed method's performance.

---

> > > ### Author Response · Authors · 2025-08-06
> > >
> > > The Neurips rebuttal instructions state that we can not include plots in the response. However, the following table shows the results averaged over each context variable for the SimpleDirection environment with stochastic transitions and rewards.
> > >
> > > |          | Context 1 | Context 2 |
> > > |----------|----------:|----------:|
> > > | Baseline |     16.50 | 16.61     |
> > > | LDR      |     37.63 | 37.81     |
> > > | CSE      |     37.86 | 37.95     |
> > >
> > > We will include the full plot of the results as the context varies in the final manuscript. Thank you again for your feedback and thoughtful remarks.

---

### Official Review · Reviewer_xxBc · 2025-07-03

**Clarity:** 3
**Significance:** 3
**Originality:** 2
**Rating:** 4
**Confidence:** 4

**Summary:**

The paper tackles zero-shot context generalization in contextual Markov decision processes (CMDPs) when only one (or very few) training context(s) are available. The main technical idea is to linearize the CMDP’s transition and reward functions around the training context, yielding the Context-Enhanced Bellman Equation (CEBE). Solving CEBE produces a Q-function that the authors prove is a first-order approximation of the true Q-function in nearby, unseen contexts. From CEBE they derive Context Sample Enhancement (CSE). This simple data-augmentation rule perturbs tuples in the replay buffer using analytic context gradients, letting standard off-policy RL algorithms train “as if” they had sampled additional contexts. Theoretical results establish that CEBE yields a first-order approximation to the Q function under smoothness assumptions, and show that an $\epsilon$-optimal policy for CEBE is $(\epsilon+\delta)$-optimal for the original CMDP. Experiments range from tabular Cliff-Walking to classic-control, a toy linear system, and Mujoco benchmarks. Across tasks, CSE matches or outperforms vanilla training and closely tracks an oracle LDR baseline.

**Questions:**

- In many simulators like MuJoCo, the gradients are not exposed. Can CSE leverage automatic differentiation through the physics engine, or will finite-difference estimates suffice?
- What is “Sample-enhanced transition and reward” in line 142? same as Context-enhanced transition and reward?
- Just out of curiosity. I understand you assume some deterministic transitions for the theory. Can you validate CSE experimentally with the case with stochastic transitions? Also, Lipschitz assumption may not hold in real-world CMDPs. Do you have any examples with that?
- Your theory gives an $\mathcal{O}(\|c-c_0\|^2)$ error, but experiments did not quantify how far from $c_0$ that bound remains tight. Could you plot $\| Q_{\text{CE}}-Q^\star \|$ versus context distance or derive an explicit radius $⁡r_{\max}$ beyond which linearization degrades?
- All experiments couple CSE with SAC. Does the technique transfer to policy-gradient methods without a replay buffer (e.g., PPO/TRPO), on-policy actor–critic, or value-based DQN in discrete domains? Even one small-scale test would clarify the generality of CSE beyond off-policy continuous-control.

**Ethical Concerns:**

["NO or VERY MINOR ethics concerns only"]

**Final Justification:**

The authors responded to my comments and questions well, along with conducting additional experiments. I will keep my score, as I feel that the weakness still remains as a limitation of this work. However, I'd be happy for this paper to be accepted!

**Limitations:**

yes

**Quality:**

3

**Strengths And Weaknesses:**

- Strengths: Theorems quantify approximation error and policy sub-optimality with proofs provided. Paper is well structured with clarity. This paper addresses the problem of generalization from scarce contexts relevant to sim-to-real robotics, meta-RL and safe deployment and shows that analytic structure can substitute for expensive domain randomization.
- Weaknesses: The analysis requires strong assumptions that the agent must observe $\delta_c R^C$ and $\delta_c T^C$ and the transition is deterministic. Real-world simulators rarely expose these gradients. The impact of this paper may depend on availability of context gradients; without them, benefits may vanish. In terms of originality, contribution is an extension to context rather than a fundamentally new paradigm.

---

> ### Author Rebuttal · Authors · 2025-07-31
>
> We thank the reviewer for their valuable comments, questions and suggestions that will improve the final manuscript.
>
> > In many simulators like MuJoCo, the gradients are not exposed. Can CSE leverage automatic differentiation through the physics engine, or will finite-difference estimates suffice?
>
> Indeed, other gradient estimation methods can improve the applicability of the method. The theory developed for CSE assumes exact gradient information. However, sufficiently accurate estimates for the gradients should suffice for CSE. Differentiable physics simulators like Brax enable automatic differentiation through the physics engine. Finite difference estimates can be used to obtain gradient estimates as well, but require that one simulates $|\mathcal{C}| + 1$ contexts. The gradients supplied by these methods can be used with CSE to improve generalization. We ran an additional experiment using automatic differentiation with the PendulumGoal environment and present the mean performance across each context sweep here:
>
> |          | Gravity | Mass    | Length  | Goal Torque |
> |----------|---------|---------|---------|-------------|
> | Baseline | -541.48 | -423.61 | -502.22 | -273.15     |
> | LDR      | -496.67 | -440.14 | -503.97 | -252.42     |
> | CSE      | -388.47 | -398.71 | -477.59 | -245.40     |
>
> In each sweep across a context variable, CSE performs better than the baseline and LDR. We will add a plot corresponding to this experiment accompanied by the above discussion in the final manuscript.
>
> > What is “Sample-enhanced transition and reward” in line 142? same as Context-enhanced transition and reward?
>
> Thank you for pointing this out. This is a typo and we will update it to “context-enhanced transition and reward” in the final manuscript.
>
> > I understand you assume some deterministic transitions for the theory. Can you validate CSE experimentally with the case with stochastic transitions?
>
> Thank you for suggesting this. We ran an additional experiment for SimpleDirection with additive Gaussian noise in the transitions and rewards. The results are similar to the SimpleDirection results with CSE performance matching LDR, both of which outperform Baseline. We will include these results in the final manuscript.
>
> In the theory for CEBE, we do not assume that the transitions are deterministic (Theorems 1 and 3 work both for deterministic and stochastic transitions). For CSE, the deterministic setting confers multiple advantages including an efficient implementation and unique method for augmenting samples. In the stochastic setting, one must directly address how the distribution $\mathcal{T}^c(s, a)$ changes as the context changes in order to develop a stochastic CSE directly from theory. Due to time and space constraints, we have not included a detailed analysis and experimental validation of CSE in the stochastic setting, but acknowledge its importance and intend to pursue it in future work.
>
> > Also, Lipschitz assumption may not hold in real-world CMDPs. Do you have any examples with that?
>
> We assume that CMDPs are Lipschitz to streamline our presentation and to give clearer proofs. This assumption is analogous to that of the Picard-Lindelöf theorem from ODEs. We expect similar results to hold under local Lipschitz assumptions, since we only need to control the Lipschitz constant to the extent that the state space gets traversed. Observe that some amount of smoothness is necessary for context generalization to be possible, since otherwise transitions and rewards could be entirely  different in nearby contexts.
>
> We have examples with transitions and rewards that are not globally Lipschitz. Consider the PendulumGoal environment, whose gradient equations are shown in Appendix F.2. In the first-order system, we can see that $\dot \phi_g$ depends on $\frac{3}{2l}$, which is not Lipschitz for all $g \in (0, \infty)$. From a physical perspective, if the length of the pendulum is low then the pendulum velocity $\phi = \frac{d \theta}{d t}$ is sensitive to the gravitational acceleration. We observe a similar behavior with the reward function for this environment, which has unbounded gradients in the context parameters. Since these gradients are unbounded in the full context space, the function is not globally Lipschitz but is locally Lipschitz. Even so, the experiments demonstrate that the policy can still generalize well in a neighborhood of the training context.
>
> > Your theory gives an $O(|c-c_0|^2)$ error, but experiments did not quantify how far from $c_0$ that bound remains tight. Could you plot $|Q_{\text{CE}} - Q^*|$ versus context distance or derive an explicit radius $r_{\text{max}}$ beyond which linearization degrades?
>
> Our theory allows one to provide quantitative bounds. Consider the Cliffwalker Environment with the reward equation specified by Eq 12 right. Note that the transition dynamics for this equation depend linearly on $T$, so the second term in the bound from Theorem 3 navishes. We find that
> $$\partial_c^2 R^c_{\text{cliff}} = -20 (1+c)^{-3} \quad \partial_c^2 R^c_{\text{goal}} = 3.75 (1+c)^{-3.5}$$
> This implies that
> $$\| \partial_c^2 R^c \| \leq \max\left( 20 (1+c)^{-3}, 3.75 (1+c)^{-3.5} \right) \leq 20$$
> With the choice of $\gamma = 0.1$ used in the experiment, we obtain
> $$\| Q_{CE}^c - Q_{BE}^c \|_\infty \leq \| c - c_0 \|^2 200 / 9 \approx 22.2 \| c - c_0 \|^2$$
>
> The radius for which the linearization is a good approximation is dependent on the structure of the reward and transition functions. As an illustrative case, consider a CMDP where the transition function is trivial: the CMDP never transitions from one state to another state. Then $DT = 0$ and $D^2T = 0$, so Theorem 2 says that $\|Q^c_{\text{CE}} - Q^c_{\text{BE}}\| <= \|c - c_0\|^2\|D^2 R\|_{\infty}$ which is the second-order Taylor expansion of the reward function. In general, we expect the radius of the approximation around a point (s, a, c) to scale directly with $\|D^2 R(s, a, c)\|^{-1}$ and $\|D^2 T(s, a, c)\|^{-1}$.
>
> > All experiments couple CSE with SAC. Does the technique transfer to policy-gradient methods without a replay buffer (e.g., PPO/TRPO), on-policy actor–critic, or value-based DQN in discrete domains? Even one small-scale test would clarify the generality of CSE beyond off-policy continuous-control.
>
> We also perform experiments with DQN in the CartGoal experiment in Appendix C.2. This environment has a discrete action space and continuous state space. We note that CSE requires a continuous state space so that $\partial_c T$ is well-defined.
>
> While we present CSE mainly in the context of SAC and do an experiment with DQN in the appendix, it should work more generally with other algorithms. Importantly, CSE is derived from the CEBE. The CEBE estimates the value of the next state in the perturbed context by bootstrapping on the Q function. Algorithms which leverage bootstrapping should work similarly. Issues may arise when using $n$-step returns for estimating returns as repeated linear approximations will accumulate error for large $n$.
>
> We will include this discussion about the algorithms in the appendix of the final manuscript.

---

> > ### Comment · Reviewer_xxBc · 2025-08-05
> >
> > I'd like to thank the authors for addressing my questions and conducting additional experiments. I really appreciate that. I will keep my score.

---

> > > ### Comment · Area_Chair_ox5M · 2025-08-06
> > > **Reviewer response details needed**
> > >
> > > Hello,
> > >
> > > The last step in the reviewing process is to process the updates from the authors that are key in clearing up final issues to ensure papers get a fair treatment. Please provide more context. Are all concerns addressed? To what extent? Any other thoughts or a summary of your stance would be good.
> > >
> > > - AC

---

### Decision · Program_Chairs · 2025-09-17

**Decision:**

Accept (poster)

**Comment:**

The reviewers do point out a number of point related to the paper.

1. The method relies on having direct access to the context and to the derivates of the reward and transition function wrt. the context. These assumptions are quite limiting as to the applicability of the method, and hence it would be beneficial to discuss whether they can be mitigated through further approximations, or whether they are fundamental to the method.
2. However the reviewers agree that the paper has some interesting components. The rebuttal was helpful to increase confidence in the reviewers understanding of the paper.
3. Future work should consider more complex environments that will require larger context. And real world environments where gradients and rewards are not easy to access.